# Explaining the high skill of Reservoir Computing methods in El Niño prediction

Francesco Guardamagna[1,2], Claudia Wieners[1,2], and Henk A. Dijkstra[1,2]

[1]Institute for Marine and Atmospheric research Utrecht, Department of Physics, Utrecht University, Utrecht, the Netherlands
[2]Center for Complex Systems Studies, Utrecht University, Utrecht, the Netherlands

**Correspondence:** Francesco Guardamagna <f.guardamagna@uu.nl>

**Abstract.** Accurate prediction of the extreme phases of the El Niño Southern Oscillation (ENSO) is important to mitigate the socioeconomic impacts of this phenomenon. It has long been thought that prediction skill was limited to a 6 months lead time. However, Machine Learning methods have shown to have skill at lead times up to 21 months. In this paper we aim to explain for one class of such methods, i.e. Reservoir Computers (RCs), the origin of this high skill. Using a Conditional Nonlinear Optimal Perturbation (CNOP) approach, we compare the initial error propagation in a deterministic Zebiak-Cane (ZC) ENSO model and that in an RC trained on synthetic observations derived from a stochastic ZC model. Optimal initial perturbations at long lead times in the RC involve both sea surface temperature and thermocline anomalies which leads to a decreased error propagation compared to the ZC model, where mainly thermocline anomalies dominate the optimal initial perturbations. This reduced error propagation allows the RC to provide a higher skill at long lead times than the deterministic ZC model.

## 1 Introduction

The El Niño Southern Oscillation (ENSO) phenomenon, driven by ocean-atmosphere interactions in the tropical Pacific, is one of the biggest sources of interannual climate variability (Neelin et al., 1998). The full ENSO cycle shows an irregular period of 2-7 years. During its warm (El Niño) and cold (La Niña) phases, ENSO strongly affects the climate all over the globe through well-known teleconnections (McPhaden et al., 2006), increasing the incidence of extreme weather events like global droughts (Yin et al., 2022) and tropical cyclones (Wang et al., 2010). ENSO can therefore have a substantial impact on the worldwide economy (Liu et al., 2023a), and accurate and reliable forecasts are necessary to mitigate its socioeconomic consequences.

For this reason, ENSO modeling and forecasting have been a central topic of extensive research, which thanks to the contribution of the Tropical Ocean–Global Atmosphere program, led to the development of a complete hierarchy of models. This hierarchy includes conceptual models (Jin, 1997; Suarez and Schopf, 1988; Takahashi et al., 2019; Timmermann et al., 2003), intermediate complexity models (Zebiak and Cane, 1987; Battisti and Hirst, 1989) and Global Climate Models (Planton et al., 2021). Many of these classical dynamical models can reasonably forecast ENSO up to a lead time of 6 months, with a correlation between predictions and observations larger than 0.5 (Barnston et al., 2012), but their skill rapidly decreases for longer lead times.

In recent years, the application of Machine Learning (ML) techniques for predicting ENSO has significantly advanced (Bracco et al., 2024). Ham et al. (2019a) showed that Convolutional Neural Networks (CNNs) trained with CMIP5 and reanalysis data could obtain reasonable skill up to lead times of about 17 months. Hu et al. (2021) advanced the CNN approach by integrating dropout and transfer learning with a residual CNN, obtaining a good performance up to a lead time of 21 months. Long Short-Term Memory (LSTM) networks, able to exploit the temporal dynamics present in the training data, have also been successfully applied to ENSO forecasting (Xiaoqun et al., 2020). More recent studies have combined LSTM with other methods like Graph Neural Networks (Jonnalagadda and Hashemi, 2023), CNNs (Mahesh et al., 2019) and AutoEncoders (Jonnalagadda and Hashemi, 2023) to create hybrid models boosting the performance, as they are able to capture both the spatial and temporal dynamics present in the data. Reservoir Computer (RC) methods, a special class of Recurrent Neural Networks (RNNs), have shown optimal performance in predicting ENSO (Hassanibesheli et al., 2022). The RC offers a good balance between performance and model simplicity, which enhances explainability and facilitates analysis of model predictions. Moreover, like other RNN-based models, the RC offers the possibility of generating a self-evolving system that does not rely on external inputs (Guardamagna et al., 2024). This characteristic is crucial to understanding the internal dynamics of the RC and the evolution of errors over time during forecasting.

All these new tools provide more accurate forecasting skills than classical dynamical models, especially for longer lead times, and seem to be able to circumvent the "Spring Predictability Barrier" (SPB). The SPB (Webster and Yang, 1992; Lau and Yang, 1996) has been identified and documented across all the ENSO's dynamical models hierarchy from conceptual models (Jin and Liu, 2021a, b; Jin et al., 2021) to comprehensive GCMs (Duan and Wei, 2013). In particular, in the intermediate-complexity Zebiak and Cane (ZC) model (Zebiak and Cane, 1987), the SPB has been rigorously studied and quantified using the Conditional Nonlinear Optimal Perturbation (CNOP) framework (Mu et al., 2007). This tool has been applied to investigate the sensitivity of the ZC model to both initial conditions (Duan et al., 2013b) and model parameters (Yu et al., 2014) uncertainties. Thus, the ZC model is an excellent testbed to analyze why ML algorithms can have skill beyond the SPB, providing good predictions even when initialized during boreal spring.

In this paper, we aim to explain the good performance of RC methods in ENSO prediction. Specifically, we will compare the evolution of optimal initial perturbations, determined using the CNOP approach, between the RC (trained with synthetic observations from the stochastic ZC model) and the deterministic ZC model. In section 2, we shortly describe the ZC model and the CNOP technique, focusing on the changes introduced to adapt them to our analysis; in addition, the RC approach is briefly presented. In section 3 we first assess the performance of the RC, and then present results of the CNOP analysis for both the RC approach and the ZC model. A summary and discussion of the results follows in section 4.

## 2 Models and Methods

### 2.1 Zebiak and Cane (ZC) model

The ZC model is an intermediate complexity ENSO model that describes the evolution of anomalies with respect to a prescribed seasonal mean climatological state across the tropical Pacific. The state vector of this model consists of two-dimensional fields

of sea surface temperature, thermocline depth, oceanic and atmospheric velocities, and the atmospheric geopotential. For a complete description of the model's components and equations, we refer the reader to Zebiak and Cane (1987). We will use both the original deterministic ZC model and a stochastic ZC model following the approach described in Roulston and Neelin (2000). In this stochastic version, only noise in the zonal wind-stress field is applied as follows. First, a linear regression model relating SST anomalies and surface zonal wind-stress anomalies was constructed empirically from observations using the ORAS5 dataset (Copernicus Climate Change Service, 2021) over the period between 1961 and 1991 with a time step of 10 days (corresponding to the ZC model time step). Next, the variability explained by this linear model was subtracted from the total zonal wind-stress field to obtain the residual zonal wind-stress anomalies. The first EOF of this residual (Fig. A1) shows a strong component over the eastern Pacific. In Feng and Dijkstra (2017), the first two EOFs were included, where the second EOF captures the westerly wind bursts, but to keep the spatial noise structure simple, we only included the first EOF. Finally, the principal component (PC) related to the first EOF was fitted to a first-order autoregressive model:

$$x_{t+1} = ax_t + b\epsilon_t, \tag{1}$$

where $\epsilon_t$ is a white noise term following a Gaussian distribution with zero mean and unit variance ($\epsilon_t \sim N(0,1)$), while $a$ and $b$ are the fitted parameters. This fitted first-order autoregressive model was used during integration to generate a different (random) zonal wind-stress anomaly pattern at each time step.

There is still a debate on whether the Pacific climate state is in a subcritical or supercritical regime (Kessler, 2002; Guardamagna et al., 2024). This distinction hinges on whether ENSO variability is a damped oscillation excited by stochastic forcing (subcritical) or occurs as a sustained oscillation or limit cycle (supercritical). In the supercritical case, ENSO behavior is strongly influenced by nonlinearities, which arise from three main sources in the ZC model: heat advection, wind stress anomalies, and subsurface water temperature variations (Duan et al., 2013a). Given this ongoing debate, we study here both regimes, which can be easily distinguished in the ZC model by varying a single parameter. Following Tziperman et al. (1994), we use a parameter $r_d$ in the drag coefficient $C_d = r_d C_d^0$, where $C_d^0$ is the standard value in the ZC model. Given the zonal and meridional wind velocities $\mathbf{u_a} = (u_a, v_a)$, the ZC model computes the wind stress $(\tau_x, \tau_y)$ acting on the ocean surface according to the bulk formula:

$$(\tau_x, \tau_y) = \rho_{air} r_d C_d^0 |\mathbf{u_a}|(u_a, v_a), \tag{2}$$

where $\rho_{air}$ is the air density, and $r_d = 1$ in the original model configuration (Zebiak and Cane, 1987). With increasing $r_d$, the ZC model generates a larger wind-stress response to sea surface temperature anomalies, intensifying the coupling strength between ocean and atmosphere.

In the deterministic version of the ZC model, an initial anomaly on the seasonal background state rapidly decays for $r_d = 0.79$ (Fig. B2a). In contrast, for $r_d = 0.8$, ENSO variability occurs as a periodic solution with a $\sim 4\,years$ period (Fig. B2b). Hence, the Hopf bifurcation bounding the two regimes is located between $r_d = 0.79$ and $r_d = 0.8$; here we choose $r_d = 0.77$ as a value in the subcritical regime and $r_d = 0.9$ in the supercritical regime. When noise is introduced, the ZC model's ENSO is phase-locked in the winter season (Fig. A3) for both $r_d = 0.77$ and $r_d = 0.9$. The SPB is identified with the initial month

corresponding to the fastest decrease in autocorrelation in eastern Pacific SST anomalies (Jin and Liu, 2021a). According to this definition, the ZC model shows a clear SPB in May for both $r_d = 0.77$ and $r_d = 0.9$ (Fig. A4). All these aspects make the ZC model a good testbed for understanding why the RC can circumvent the SPB, both in the subcritical and supercritical regime.

## 2.2 Reservoir Computer

Although the procedure to generate a RC has been well described elsewhere (Pathak et al., 2018), we briefly summarise the approach here, also introducing our notation. Given an input signal $u(n) \in R^{N_u}, n = 1, .., N_t$, where $N_t$ is the total number of time steps and a given output signal $y^{target}(n) \in R^{N_y}$, the RC has to learn how to estimate an output signal $y(n) \in R^{N_y}$ as similar as possible to $y^{target}(n)$. To do that during the training procedure, an error measure $E(y, y^{target})$ is minimized, for which we choose a common measure for regression problems: the Mean Squared Error (MSE) defined by

$$100 \quad \mathrm{E}(y, y^{target}) = \frac{1}{N_y} \sum_{i=1}^{N_y} \left( \frac{1}{N_t} \sum_{n=1}^{N_t} (y_i(n) - y_i^{target}(n))^2 \right). \tag{3}$$

Before the training procedure, the input data $u(n)$ are nonlinearly expanded into a higher dimensional so-called reservoir space, generating in this way a new signal $x(n) \in R^{N_x}$. This new representation of the data also contains temporal information and is based on the following update equations:

$$\tilde{x}(n) = \tanh \left( W^{in} u(n) + W x(n-1) \right), \tag{4a}$$

$$x(n) = (1 - \alpha) x(n-1) + \alpha \tilde{x}(n), \tag{4b}$$

where the hyperbolic tangent ($\tanh$) is applied component wise. Including a non-linear activation function such as $\tanh$ in the update equations enables the RC to estimate nonlinear relationships among the input variables, in contrast to less sophisticated models like the Linear Regressor, which can only capture linear relationships. This gives the RC an advantage in scenarios where nonlinearities play a significant role. The two matrices $W^{in} \in R^{N_x \times N_u}$ and $W \in R^{N_x \times N_x}$ are generated randomly
according to chosen hyperparameters. The non-zero elements of $W$ and $W^{in}$ are sampled from a uniform distribution over the range $[-a, a]$. The sparse matrix $W$ derives from a random network with mean degree $< k >$, while $W^{in}$ is a dense matrix. The quantity $\alpha \in (0, 1]$ in (4b) is the leaking rate. The output layer is defined as $y(n) = W^{out} x(n)$, where $W^{out} \in R^{N_y \times N_x}$, and during the training procedure only the weights of $W^{out}$ are estimated by minimising $E(y, y^{target})$ through a linear regression procedure. We use a ridge regression to avoid overfitting, leading to the loss function $\mathcal{L}$:

$$110 \quad \mathcal{L}(W^{out}) = E(y, y^{target}) + \epsilon \sum_{i=1}^{N_y} \sum_{j=1}^{N_x} (W_{i,j}^{out})^2. \tag{5}$$

The hyperparameters are given by the dimension of the reservoir ($N_x$), the spectral radius of the matrix $W$ ($\rho$), the sparsity of $W$'s connections $< k >$, the input scaling $a$ and the leaking rate $\alpha$. Given an input sequence $u(n) = y^{target}(n)$, the RC is trained by determining $W^{out}$ from the sequence $y(n) = u(n+1) = y^{target}(n+1)$, using the loss function (5).

After training, the RC can be transformed into an autonomous evolving dynamical system to be used for prediction (Pathak et al., 2018). Thereto feedback connections between the outputs at time step $n$ and the inputs at the subsequent time step are introduced. In this way, a model is generated that autonomously evolves in time, according to

$$
\begin{aligned}
x(n+1) &= (1-\alpha)x(n) + \alpha \tanh\left(Wx(n) + W^{in}u(n+1)\right), & \text{(6a)} \\
u(n+1) &= y(n) = W^{out}x(n), & \text{(6b)}
\end{aligned}
$$

where $x(n)$ and $x(n+1)$ are the reservoir states at time step $n$ and $n+1$, while $y(n)$ is the output at time step $n$ and $u(n+1)$ is the input at the subsequent time step $n+1$. This property of the RC allows us to make predictions similar to classical dynamical systems. Consequently, we can study how an initial perturbation evolves in the RC.

In the results below, the input vector $u$ consists of the following feature variables: the NINO3 index, the thermocline depth anomalies $h_W$ and $h_E$ averaged over the regions 5°N-5°S $\times$120°E-180°E, and 5°N-5°S $\times$180°E-290°E, respectively and the zonal surface wind-speed anomalies $\tau_C$ averaged over the area 5°N-5°S $\times$ 145°E-190°E. Instead of directly using zonal surface wind-stress anomalies, zonal surface wind-speed anomalies have been used as a proxy. The two variables are inherently correlated through the bulk formula (2) and therefore convey similar information. However, a key distinction arises from how noise is introduced in the ZC model, specifically in the form of random zonal wind-stress anomalies. This leads to random local fluctuations in the zonal wind-stress signal, which are inherently difficult for the RC to predict and reproduce. In contrast, the surface wind-speed anomaly signal is smoother and more predictable, making it easier for the RC to learn and generalize efficiently.

In addition to the previous variables, a sine signal with a 12-month period was included to represent the seasonal cycle, such that $N_u = 5$. Although a combination of sine and cosine signals is required to identify uniquely each month of the year, we found that including both made little difference in performance. Therefore, to minimize the number of input variables and reduce the complexity of the learned function, we decided to use only the sine signal. The output vector consists of the same variables as in the input except for the sine signal, hence $N_y = 4$. In self-evolving mode, the sine signal encoding the seasonal cycle is provided as an external input rather than generated directly by the RC.

## 2.3 CNOP computation

Our implementation of the CNOP methodology follows the one described by Duan et al. (2013b). Let $M_{t_0,t}$ be the propagator of a nonlinear model from initial time $t_0$ to a chosen end time $t_e$. We indicate $v_0$ as the initial perturbation superimposed on the model's background state $V_0$ at time $t_0$. For a selected norm $||.||$, an initial perturbation $v_{0\delta}$ is defined as a CNOP if and only if:

$$
\begin{aligned}
J(v_0) &= ||M_{t_0,t_e}(V_0 + v_0) - M_{t_0,t_e}(V_0)||, & \text{(7a)} \\
J(v_{0\delta}) &= \max_{C(v_0)\leq\delta} J(v_0), & \text{(7b)}
\end{aligned}
$$

where $C(v_0)$ is the constraint condition and $M_{t_0,t}(V_0)$ represents the model state at time $t$ when the integration starts from the background state $V_0$ at time $t_0$. In Duan et al. (2013b), an initial perturbation is applied to all the grid points over the tropical

area and the constraint condition to the initial perturbation amplitude $C(v_0)$ is defined as:

$$C(v_0) = \sqrt{\sum_{i,j} [(w_T^{-1} T'_{i,j})^2 + (w_h^{-1} h'_{i,j})^2]},  \tag{8}$$

where $T'_{i,j}$ and $h'_{i,j}$ are the initial sea surface temperature anomalies (SSTA) and thermocline depth anomalies, respectively, at grid point $(i,j)$. The weights $w_T = 2°C$ and $w_h = 50$ m represent the characteristics scale of SST and thermocline depth anomalies, respectively.

As mentioned in section 2.2, the RC is trained using a limited feature vector. To ensure a fair comparison of CNOPs between those of the RC and the ZC models, the tropical area of the ZC model is divided into boxes and uniform perturbations are applied over those boxes. Specifically, we apply a uniform SSTA perturbation $T'_E$ over all the grid points in the NINO3 area (5°N-5°S × 210°E-270°E), a uniform thermocline depth perturbation $h'_W$ to all the grid points in the area 5°N-5°S ×120°E-180°E and a uniform thermocline depth perturbation $h'_E$ to all the grid points in the area 5°N-5°S × 180°E-290°E. The constraint condition can then be written as:

$$C(v_0) = \sqrt{(w_T^{-1} T'_E)^2 + (w_h^{-1} h'_E)^2 + (w_h^{-1} h'_W)^2}.  \tag{9}$$

For both the RC and the ZC model, the objective function $J(v_0)$ in (7) has been defined as the Root Squared Error (RSE) between the perturbed and background trajectories. Specifically if we define the NINO3 index value at time $t$ when the integration start from the initial state $V_0$ as NINO3$(t, V_0)$, the objective function $J(v_0)$ is defined as:

$$J(v_0) = \sqrt{\sum_{t=t_0}^{t=t_N} (\text{NINO3}(t,(V_0 + v_0)) - \text{NINO3}(t, V_0))^2},  \tag{10}$$

where $t_N = t_e$. To solve the optimization problem associated with determining the CNOP, we use the gradient-free Cobyla optimization algorithm (Powell, 1994). Since the Cobyla algorithm starts its optimization process from a random initial guess, we always perform 10 different realizations starting from 10 different initial guesses to select the CNOPs that shows the largest error propagation according to the value of $J(v_0)$; a detailed description of the COBYLA algorithm is reported in appendix B.

## 3   Results

In the results section, we will first explain the training and validation of the RC (section 3.1), demonstrate the forecasting skills of the RC (section 3.2), also demonstrating the importance of the zonal surface wind velocity anomalies as a training variable. Next, in subsection 3.3, we present the results of the CNOP analysis for both the RC and deterministic ZC models.

### 3.1   Training and Validation of the RC

For both subcritical ($r_d = 0.77$ ) and supercritical ($r_d = 0.9$) regimes we first performed a simulation of 1000 years with the stochastic ZC model using a time step of 10 days. We will refer to these data as 'synthetic observations'. The NINO3 amplitudes of the supercritical case (Fig. 1b) are, as expected, about a factor 2 larger than those of the subcritical case

(Fig. 1a). As mentioned in section 2.2., the 12 months period sine signal and the feature vector components $h_W$, $h_E$, $\tau_C$, and NINO3 (extracted from the synthetic observation time series) are used to train the RC. To investigate the effect of $\tau_C$ on the performance of the RC, we also trained a second RC using only $h_W$, $h_E$, NINO3 and the sine signal. Before training the NINO3 and both $h_W$ and $h_E$ have been normalized by $w_T = 2°C$ and $w_h = 50m$, respectively. From the total 1000 years of synthetic observations, the first 300 years were discarded to avoid capturing any initial transient behavior. The next 500 years were used for training and validation (300 years for training and 200 years for validation), and the last 200 years were used for testing, ensuring an independent evaluation of the RC model performance. The training of the RC was described in section 2.2, where given an input sequence $u(n) = y^{target}(n)$, $W^{out}$ is determined from the sequence $y(n) = u(n+1) = y^{target}(n+1)$, using the loss function (5).

To determine the performance of the RC, we use the RC in self-evolving mode (section 2.2) to make predictions using a time step of 10 days. When we let the RC self evolve, the only external information we provide is the value of the sine signal representing the current month of the year. All the other variables (NINO3, $h_E$, $h_W$, and $\tau_C$ when the latter is included as a training variable) are directly produced by the output of the RC and are not provided as external information during prediction. To evaluate the RC's performance over the entire 200 years of validation trajectories for different lead times, we adopt a rolling approach. For each time step $t(n)$ in the validation data set, a RC trajectory with a specific lead time is generated. The final values of each trajectory, corresponding to the lead time of interest, are then concatenated to form a complete 200-years trajectory, say $y_{full}$. Before performing inference, we always determine the internal RC state using 5 years of data prior to the time step $t(n)$. Discarding the initial $x(n)$ RC states for $0 \leq n \leq n_{transient}$ is a common practice in Reservoir Computing. This step is necessary to mitigate the impact of initial transients caused by the arbitrary initialization of the reservoir state, typically set to $x(0) = 0$, or randomly initialized. In our case, we set $x(0) = 0$. This initialization creates an artificial starting state that is unlikely to recur once the reservoir dynamics stabilize. A warm-up period is, therefore, necessary to allow the RC to reach a stable dynamical regime. The length of this warm-up period depends on the RC's memory capacity and the specific learning task. Based on our experiments, we found that a 5-year warm-up period is sufficient to stabilize the reservoir dynamics and eliminate the effects of initial transients. As a result, the reservoir states corresponding to the 5 years of data preceding the time step $t(n)$ are used to initialize the RC internal state and then discarded. In our notation, this means discarding the $x(n)$ reservoir states for $t(n) - n_{transient} \leq n \leq t(n)$, where $n_{transient} = 180$, given our 10-day time step.

To identify the best set of hyperparameters, a separate validation procedure was conducted for each regime ($r_d = 0.77$ and $r_d = 0.9$) and for each set of training variables (including and excluding the zonal surface wind speed anomalies) using a Bayesian search. For each hyperparameter set, the RC model's 18-month lead time predictions were evaluated using the root mean square (RMS) error computed including all feature variables in $y_{full}$. This latter was done to ensure that the RC model could replicate the synthetic observations for all variables of interest rather than simply replicating the NINO3 index. Among all the different hyperparameters, the reservoir dimension $N_x$ is one of the most significant for the RC's performance (Lukoše-vičius (2012); Verstraeten et al. (2007)). Increasing $N_x$ expands the reservoir's state space, allowing for a richer and more complex high-dimensional representation of the input signal $u(t)$ (see Section 2.2). Additionally, larger $N_x$ values increase the reservoir's memory capacity.

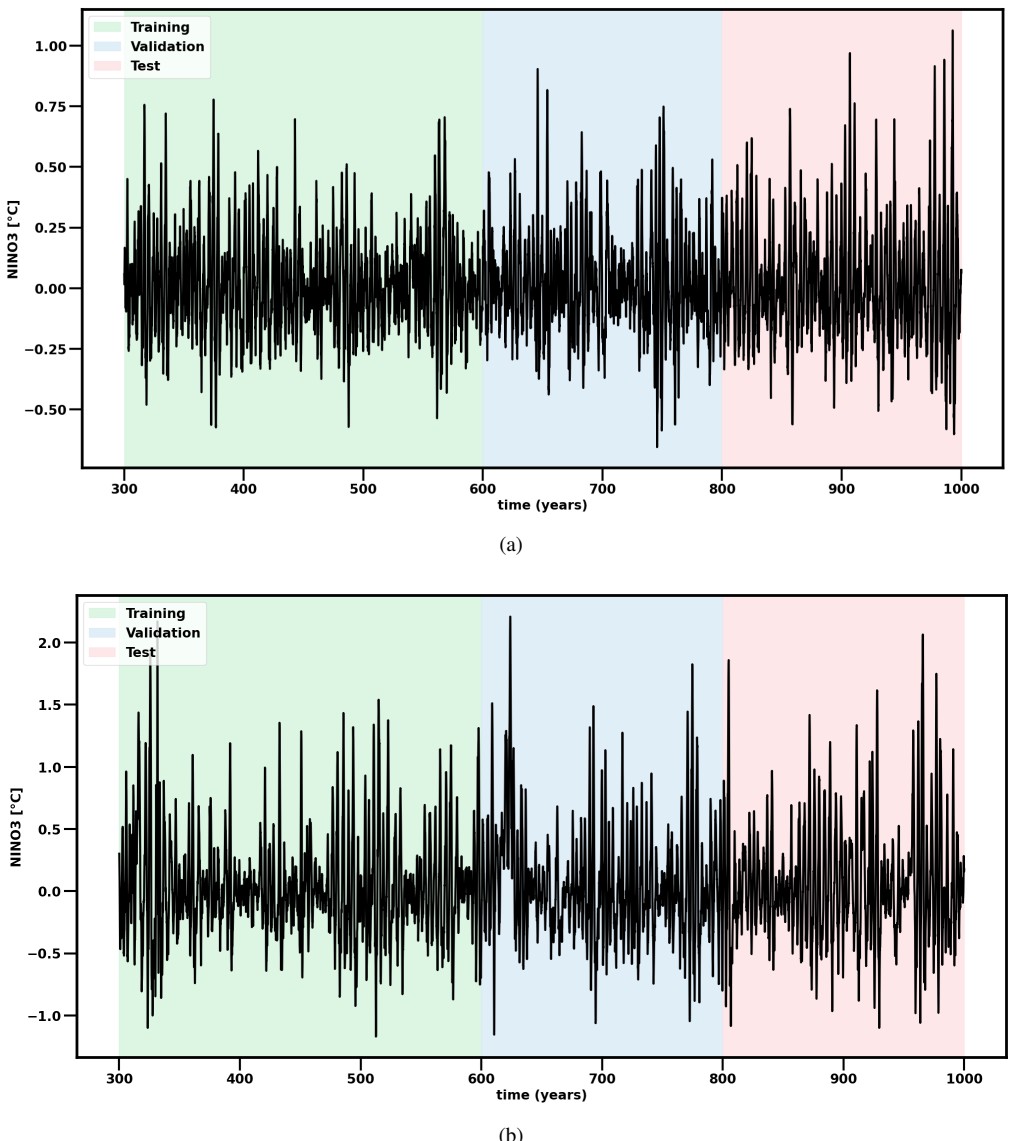

**Figure 1.** NINO3 index from the last 700 years of the stochastic ZC model simulations (synthetic observations) used to train, validate, and test the RC model: (a) $r_d = 0.77$, (b) $r_d = 0.9$.

During our experiments, the Bayesian search has consistently converged on large $N_x$ values, with a notable difference between the supercritical and subcritical regimes. In the supercritical regime, the optimal reservoir dimension is approximately 400, regardless of whether the variable $\tau_C$ is included during training. In the subcritical regime, the optimal dimension is larger, with $N_x = 476$ when $\tau_C$ is included and $N_x = 534$ when it is excluded. Table C1 in appendix C reports the optimal $N_x$ values identified for each regime and input-variables configuration, as well as the optimal values for all other RC's hyperparameters (see Section 2.2).

After validation, we evaluated the RC model's performance on the 200-year test set using these best hyperparameter sets, as described next.

## 3.2 RC performances

Figure 2 presents the mean and standard deviation of the Anomaly Correlation Coefficient (ACC) of 50 different RC's prediction trajectories and the target NINO3 index from the 200-years test dataset, computed at a monthly time step (so averaged over three model time steps). We evaluated the RC's ability to replicate the monthly NINO3 index rather than the 10-day time step index used for training, since this is the common approach for assessing the performance of ENSO forecasting models. As the reservoir is generated by random $W$ and $W^{in}$ values, each RC needs to be retrained first (using the 300 years data set) as described in section 2.2 and hence multiple RCs are used for evaluating the ACC. Again a rolling approach (as for the validation data set, see (section 3.1) was used for the test set and hence the ACC is determined using the 200-years vector $y_{full}$.

In the supercritical regime (Fig. 2b), the RC model performs better when zonal surface wind-speed anomalies $\tau_C$ are included as a training variable, though its performance is also acceptable even when $\tau_C$ is excluded. On the other hand, in the subcritical regime (Fig. 2a), the RC performance for longer lead times (9 to 18 months) improves when $\tau_C$ is excluded during training. In this regime ENSO is primarily driven by atmospheric noise, introduced in the Zebiak and Cane model in the form of random zonal wind-stress burst (see section 2.1). When the model is initialized from ENSO neutral conditions, optimal atmospheric noise patterns can trigger transient growth of perturbations, provided the initial conditions are favorable. Conversely, if a perturbation is already developing, subsequent noise patterns can either reinforce or damp its evolution. The variable $\tau_C$ is therefore particularly useful for predicting the short-term variability of ENSO, as it provides critical information about external forcing that influences early perturbation dynamics. Accordingly in the subcritical regime the RC achieves better performances at shorter lead times (3-6 months), when $\tau_C$ is included. At longer lead times (9-18 months), improved predictive performances requires the RC to rely more on system internal dynamics rather than the short-term influence of stochastic noise. Including $\tau_C$ during training can lead to overfitting, causing the model to focus excessively on short-term noise patterns instead of learning the internal system dynamics. In the supercritical regime, nonlinearities play an essential role (see Section 2.1). In the ZC model, these nonlinearities arise from three main sources: heat advection, wind stress anomalies, and water temperature variations (Duan et al. (2013a)), all of which influence the evolution of ENSO. This characteristic is reflected in the RC model's performance, which exhibits improved predictive skill at both short and long lead times when $\tau_C$ is included during training, underscoring the importance of the nonlinear effects introduced by this variable in this regime.

Overall, the RC performs better in the supercritical regime, achieving an ACC of 0.8 at a 12-month lead time when zonal surface wind speed anomalies are included during training. In contrast, in the subcritical regime, the RC model achieves an ACC of 0.75 at a 12-month lead time when $\tau_C$ is excluded during training.

To better appreciate the performance of the RC model, we also compared it with a simple Linear Regressor as a benchmark; results are also included in Fig. 2. The comparison between the LR and RC reveals that the performance improvement achieved by adopting the RC model is not drastic. However, the results still demonstrate a clear and consistent advantage in using the RC, both in the subcritical and supercritical regimes, and regardless of whether the variable $\tau_C$ is included during training.

This improvement stems from the nonlinear activation function used in the RC (see section 2.2), which enables it to capture nonlinear relationships between input variables something the LR model, limited to linear approximations, cannot achieve.

This advantage is particularly clear in the supercritical regime, where the RC model provides a more significant performance increase compared to the subcritical regime. This is expected, as nonlinearities play a more prominent role in the supercritical regime (see Section 2.1). Moreover, the relatively small performance gap between the LR and the RC can be attributed to the ZC model being a model of intermediate complexity in which ENSO is a weakly nonlinear phenomena (e.g. all wave dynamics in ocean and atmosphere is linear in the model). The ZC model's data exhibit simpler dynamics than real-world observations

or simulations with more complex General Circulation Models (GCMs). In such cases, the performance advantage of the RC over the LR is expected to be more pronounced.

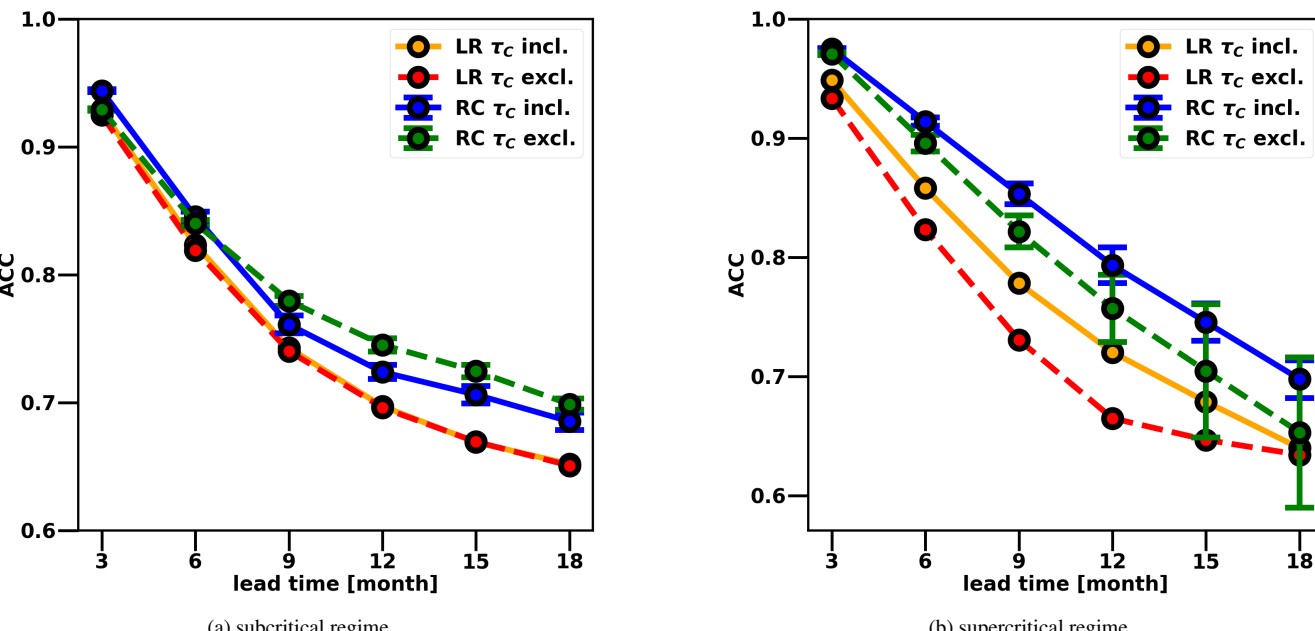

(a) subcritical regime          (b) supercritical regime

**Figure 2.** Mean and standard deviation of the Anomaly Correlation Coefficient (ACC) of 50 different RC model realizations and the 200-years synthetic observations for the NINO3 index, computed at a monthly time step. Results are shown for the 2 regimes: (a) subcritical ($r_d = 0.77$) and (b) supercritical ($r_d = 0.9$), with zonal surface wind speed anomalies ($\tau_C$) either included or excluded during training. Results from the Linear Regressor are also included for comparison

The ability of the RC model to mitigate the SPB is demonstrated in Fig. 3. This figure presents the normalized mean absolute error (MAE) between the median NINO3 of 50 different RC's predictions and the corresponding target values from the synthetic observations test dataset (see section 3.1) at various lead times and for both the RC initialized before the SPB in

March, April, and May, and after the SPB in September, October, and November. As a comparison benchmark, the normalized MAE for the Linear Regressor (LR) predictions is also included for the same initialization months. Additionally, to ensure a fair comparison between the subcritical and supercritical regimes, all RC and LR predictions and the corresponding target values

have been normalized by the standard deviation of the 200-years synthetic observations test dataset (0.47 for the supercritical regime and 0.24 for the subcritical regime) before computing the MAE. In Fig. 3, we present results for the different input variable configurations for both the subcritical and supercritical cases. Specifically, the variable $\tau_C$ is excluded from the input variables in the subcritical regime but included in the input variables in the supercritical regime.

In both the subcritical and supercritical regimes, the RC model outperforms the LR also in terms of mean absolute error, regardless of the initialization period. However, to a certain extent, it is still affected by the SPB, which occurs in May in the ZC model (as discussed in Section 2.1). On the other hand, the RC model demonstrates a clear ability to mitigate the effects of the SPB compared to the LR. This can be most clearly seen when comparing the pre-spring initialization performance of the two models at 3-months and 6-months lead times. In the supercritical regime, with pre-SPB initialization, the RC model achieves a normalized MAE of 0.2 at 3-months lead time and 0.35 at 6-months lead time, while the LR shows a higher normalized MAE of 0.3 at 3-months lead time and 0.5 at 6-months lead time. In the subcritical regime, with pre-SPB initialization, the RC achieves a MAE of 0.34 at 3-months lead time and 0.5 at 6-months lead time, while the LR shows a MAE of 0.36 at 3-months lead time and 0.54 at 6-months lead time. In the supercritical regime, the RC shows a larger performance improvement, compared to the subcritical regime where the difference in performance with respect to the LR is less evident. Moreover, also in terms of normalized MAE, the RC performs better in the supercritical regime than in the subcritical one.

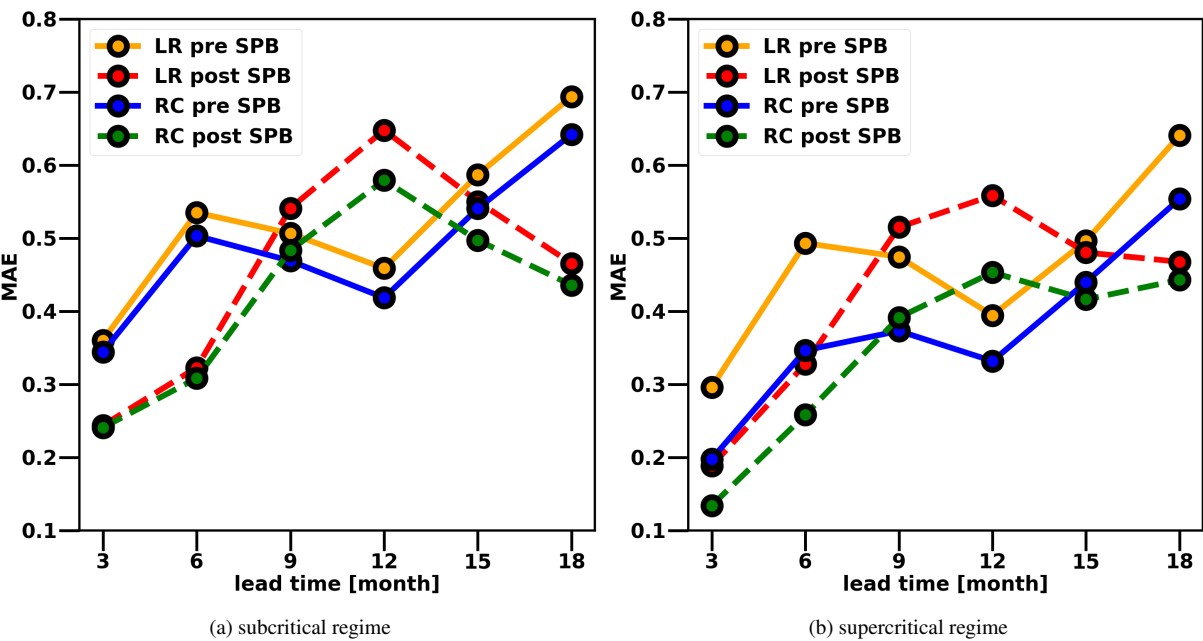

(a) subcritical regime          (b) supercritical regime

**Figure 3.** Normalized mean absolute error (MAE) between the median of 50 different RC realizations' predictions and the 200-years synthetic observation test set, for the NINO3 index computed at a monthly time step, and with the RC initialized both before (March, April, May) and after (September, October, November) the SPB: (a) $r_d = 0.77$, zonal surface wind speed anomalies excluded during training. (b) $r_d = 0.9$, zonal surface wind speed anomalies included during training. Results from the Linear Regressor (LR) are also included for comparison. Both predictions and target values have been normalized by the standard deviation of the 200 years synthetic observations test dataset (0.47 for the supercritical regime and 0.24 for the subcritical regime), before computing the MAE.

## 3.3 CNOP analysis

For both the RC model and the deterministic ZC model and for both $r_d = 0.77$ (subcritical regime) and $r_d = 0.9$ (supercritical regime), we computed the CNOPs for different lead times using the last 50 years of the 200-years synthetic observations test dataset as initial conditions (cf. section 3.1). This choice has been made to balance computational efficiency and statistical significance. The CNOP computations using the Cobyla algorithm are highly computationally expensive, and 50 years of data is sufficient to obtain statistically significant results. By selecting the last 50 years of the 200-year test period, we also ensure complete statistical independence between the training and test data. For the RC model, perturbations were directly applied to NINO3 and mean thermocline depth anomalies ($h_E$ and $h_W$). In contrast, for the deterministic ZC model, a uniform perturbation was applied over three different boxes in the Pacific for both SSTA and thermocline depth anomalies (as described in section 2.3).

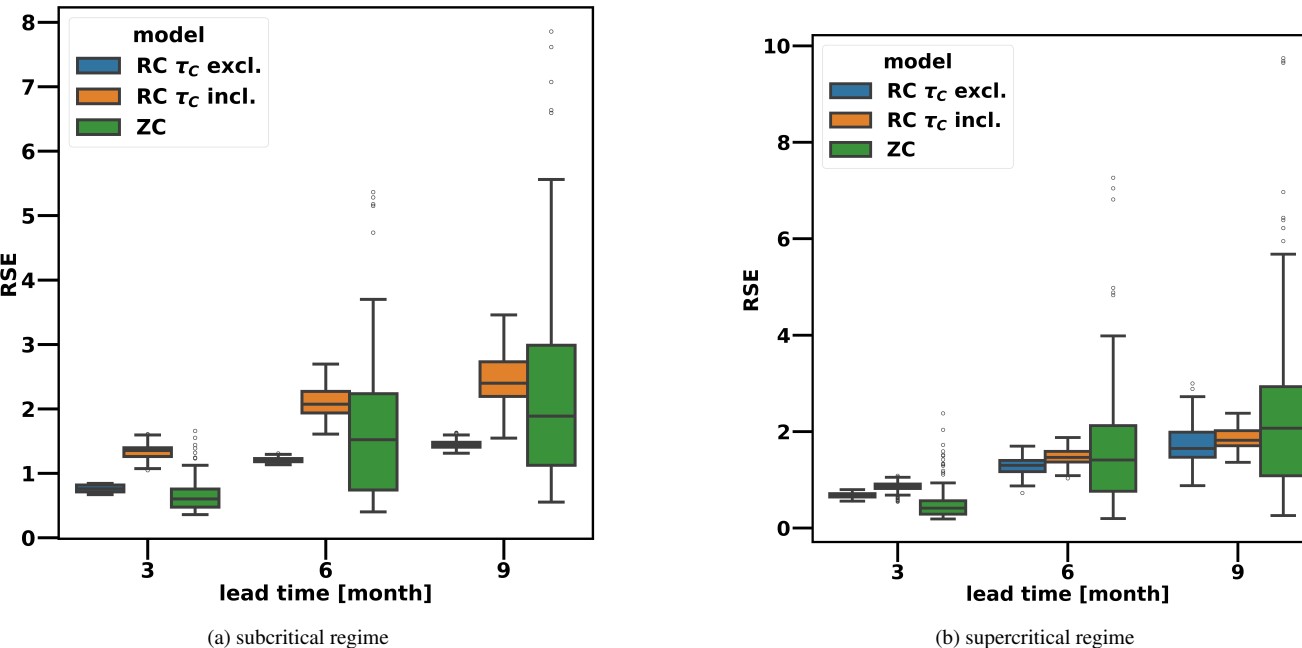

(a) subcritical regime    (b) supercritical regime

**Figure 4.** Distribution of the normalized RSE distances between the perturbed and unperturbed trajectories for different lead times when CNOPs are applied, taking as initial conditions the months [March,April,May] for (a) $r_d = 0.77$ and (b) $r_d = 0.9$. The boxes indicate the interquartile range (IQR), the range within which the central 50% of data points are located. The whiskers extend to the minimum and maximum values within 1.5 times the IQR from the first and third quartile. The central line corresponds to the median. All the RSE distances have been normalized by the standard deviation of the NINO3 index extracted from the 50 years of synthetic observations considered for the CNOP computation (0.29 for the subcritical regime and 0.56 for the supercritical regime).

Before computing the CNOPs for the RC model, we identified and saved the best-performing RC realization out of 50 for each combination of lead time, $r_d$ value, and training variables set based on the forecasting skill for the 200-years synthetic observations test period (see section 3.1). The top-performing realization (for each combination of lead time, $r_d$ value and

training variables) was then considered for the CNOP computation. This was done to avoid biases related to the random initialization of the RC. We computed the CNOPs for lead times 3, 6, and 9 months (the optimization time considered during the CNOPs computation), focusing on a single constraint value $\delta = 0.05$ and a specific forecast initialization season just before the SPB, encompassing March, April, and May. The value of $\delta$ corresponds to a maximum NINO3 perturbation of 0.1°C (section 2.3) or a maximum $h_e$ or $h_w$ perturbation of 2.5m. For the deterministic ZC model, the same procedure to compute the CNOP was used (see section 2.3). To quantify the divergence between two trajectories caused by the CNOPs, for both the RC and the ZC models, we computed the Root Square Error (RSE) distance between the perturbed and unperturbed NINO3 trajectories, as defined in [10]. This distance was used to estimate each model's sensitivity to initial condition perturbations, given a specific initial state. To make a fair comparison between the subcritical and supercritical regimes, all the RSE distances obtained have been normalized by the standard deviation of the 50 years of NINO3 synthetic observations considered for the CNOPs computation (0.29 for the subcritical regime and 0.56 for the supercritical regime).

In the supercritical regime (Fig. 4b), the RC model is more susceptible to initial perturbations at shorter lead times. However, at a 6-months lead time, the RC model's sensitivity to initial perturbations becomes, on average, smaller when $\tau_C$ is excluded during training and similar to that of the deterministic ZC model when $\tau_C$ is included during training (see Table D2). At 9-months lead time, the RC's sensitivity to initial perturbations is on average smaller for both $\tau_C$ included and excluded. At both 6- and 9-months lead times, the deterministic ZC model's sensitivity results show a much wider distribution than the RC, regardless of whether $\tau_C$ is included or excluded as a training variable. In the subcritical regime (Fig. 4a), the RC model becomes more susceptible to perturbations than the deterministic ZC model when $\tau_C$ is included as a training variable. Conversely, when this variable is excluded, the RC model shows less sensitivity to perturbations than the deterministic ZC model. This difference is likely because including $\tau_C$ as a training variable causes the RC model to learn more the noise component of synthetic observations. Since ENSO variability in the subcritical regime is highly affected by noise, including these anomalies during training leads to a system with a larger error propagation.

Previous studies (Mu et al., 2007) have quantified the SPB in the deterministic ZC model using the CNOP framework, revealing that the deterministic ZC model is particularly sensitive to initial perturbations when initialized just before the boreal spring season. Our results support this finding, showing that the deterministic ZC model exhibits a stronger sensitivity to initial condition perturbations when initialized close to the SPB than when it is initialized later in the year (see Table D1). This also holds for summer initialization (not shown) in June, July, and August, where the models show results similar to spring initialization in March, April, and May, with the RC mitigating sensitivity to initial perturbations similarly. This behavior is due to the proximity of the summer season to the SPB. The CNOP cost function evaluates the distance between the entire perturbed and unperturbed trajectories (see 10), taking all months into account. When the deterministic ZC model integration is initialized just before the SPB, the number of months affected by the SPB is maximized, and at longer lead times (6 and 9 months), we observe a pronounced increase in the sensitivity to initial conditions perturbations compared to when the model is initialized in the autumn and winter seasons. This effect is also found when comparing the sensitivities of autumn and winter initializations. Compared to an autumn-initialized trajectory, an integration initialized in winter has crossed the SPB before at

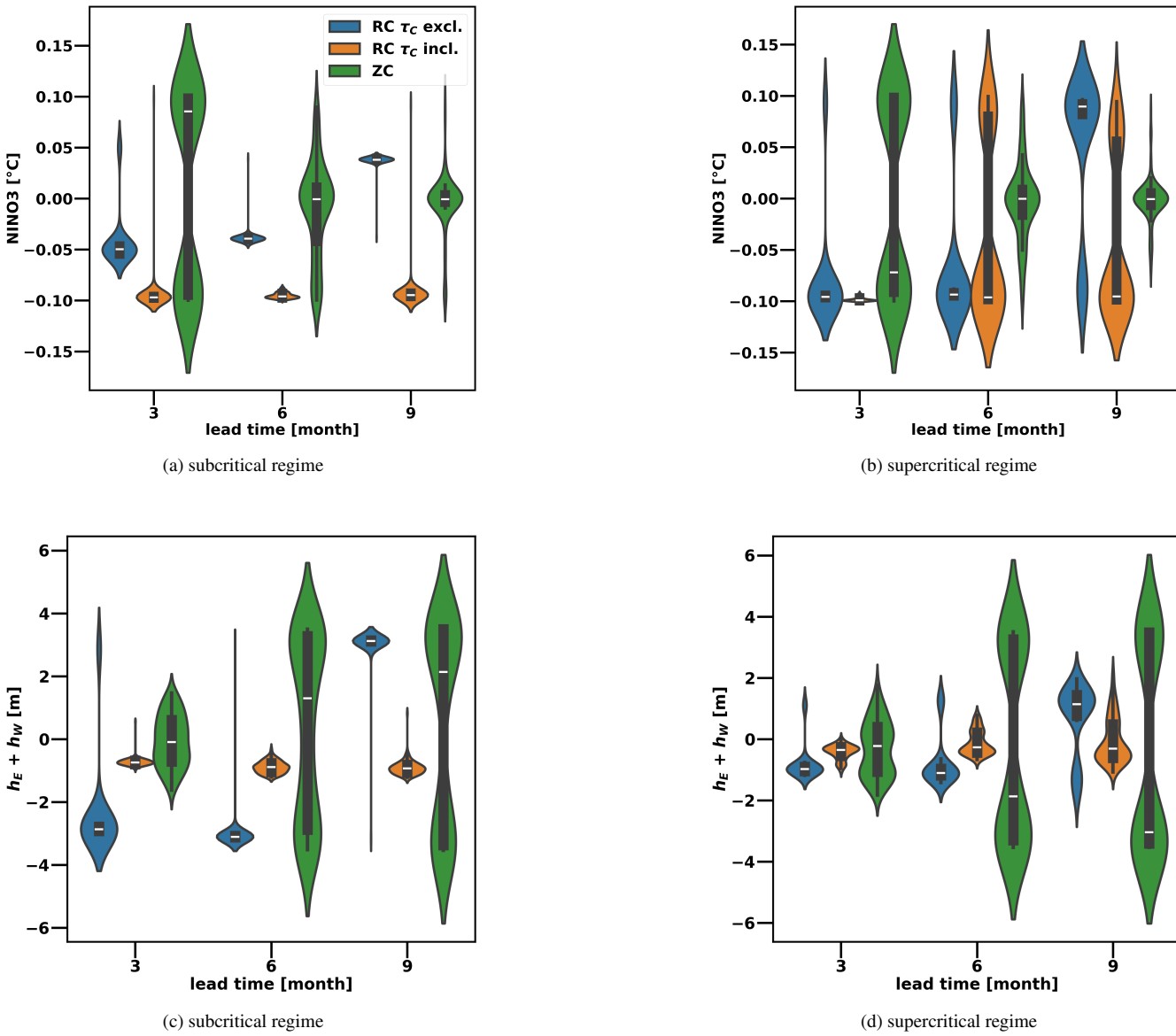

**Figure 5.** Violin plots showing the distribution of the CNOPs obtained for both the NINO3 Index and $h_E + h_W$ (sum of the thermocline anomalies for both the western and eastern Pacific). (a)-(c) $r_d = 0.77$ (b)-(d) $r_d = 0.9$. In both cases, $\delta = 0.05$, the period considered corresponds to the last 50 years of the 200 years synthetic observation test dataset, and the months [Mar, Apr, May] are taken as initial conditions.

a 9-months lead time and the sensitivity to initial perturbations for an integration initialized in winter is, on average, larger than for the autumn initialization.

On the other hand, when the RC is initialized later than the SPB, it exhibits a sensitivity to initial perturbations similar to that found when it is initialized just close to the SPB (see Table D2). As the number of months affected by the SPB increases (at 6- and 9-months lead time, with a forecast initialized in spring), the RC effectively reduces both the average sensitivity
to initial condition perturbations and the width of sensitivity results' distribution compared to the ZC model, consequently decreasing the number of events strongly sensitive to initial conditions perturbations. The only exception is the RC trained, including the variable $\tau_C$ in the subcritical regime, which consistently has a greater sensitivity to initial condition perturbations than the deterministic ZC model, for reasons already mentioned above. Moreover the inclusion of this variable decreases the performance of the RC in the subcritical regime at longer lead times (see section 3.2). These results demonstrate that the RC
model effectively mitigates the sensitivity to initial condition perturbations at long lead times (6, 9 months) when a forecast is initialized just before the SPB, compared to the ZC model, for which the spring season corresponds to the strongest sensitivity to initial perturbations. This capability explains why the RC model can reduce the effects of the SPB, delivering skillful predictions at long lead times.

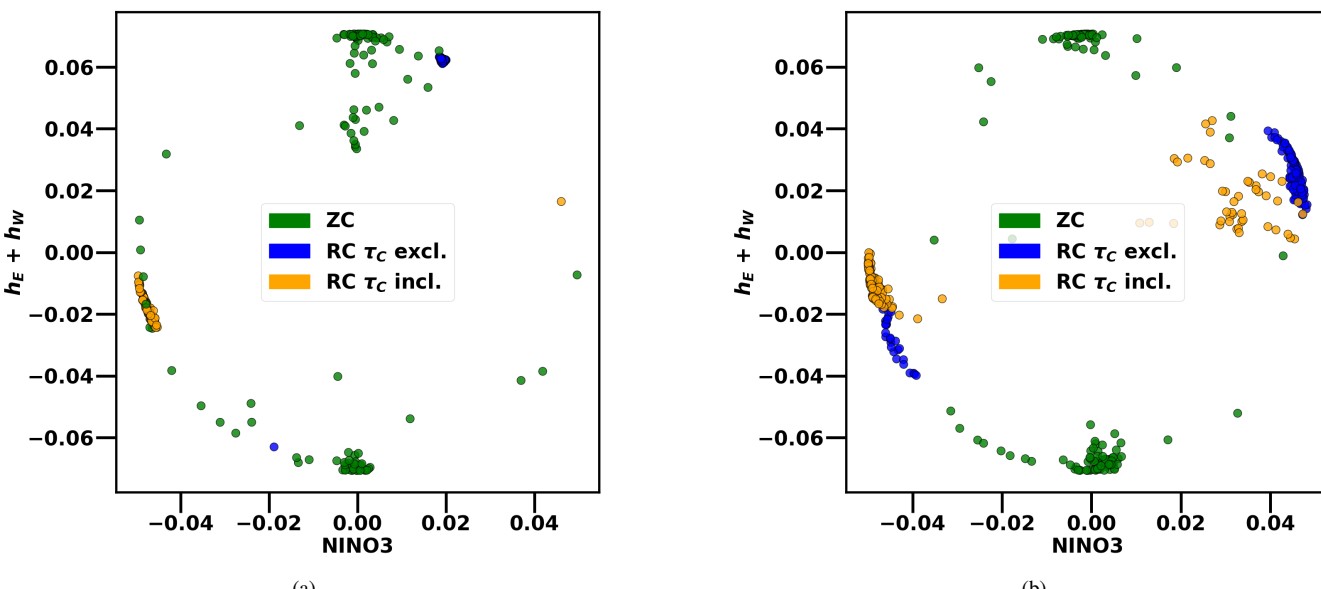

(a)                                                                 (b)

**Figure 6.** Scatter plot of the CNOPs in the normalized NINO3 index, $h_E + h_W$ anomaly plane. (a) $r_d = 0.77$ (b) $r_d = 0.9$. In both cases, $\delta = 0.05$, the period considered corresponds to the last 50 years of the 200-years synthetic observation test dataset, the months [Mar, Apr, May] are taken as initial conditions and the lead time considered is 9 months. The NINO3 index and $h_E + h_W$ anomalies have been normalized dividing by 2°C and 50m, respectively.

Figure 5 shows the estimated CNOPs for both the ZC and RC models when initialized just before the SPB in March, April,
and May. The estimated CNOPs are presented for the NINO3 index and the sum of the thermocline perturbations in the eastern and western Pacific ($h_E + h_W$). The ZC model's sensitivity to initial NINO3 perturbations decreases as the forecasting lead time increases. In contrast, perturbations to the thermocline depth become increasingly crucial for optimal perturbation growth

at longer lead times. This is true for both $r_d = 0.77$ and $r_d = 0.9$. On the other hand, the CNOPs of the RC have different behavior, both for the supercritical and subcritical regimes. The RC is sensitive to quite different initial perturbations leading to less variability in the error propagation compared to the ZC model. This is supported by Fig. 6, which shows the distribution of the CNOPs in the (NINO3, $h_E + h_W$) plane for a 9 months lead time. For visualization purposes, the initial anomalies for NINO3 and ($h_E + h_W$) have been normalized dividing by 2°C and 50m, respectively (see section 2.3). The CNOPs for the RC and the ZC models show a strongly different distribution. The RC model consistently exhibits greater sensitivity to NINO3 perturbations at both short (3 months) and long (6 and 9 months) lead times, while the ZC model shows increasing sensitivity to initial thermocline depth perturbations as the optimization time extend. In the ZC model, ENSO variability is strongly influenced by thermocline feedback (Zebiak and Cane (1987)). The RC reduces this sensitivity at longer lead times helping to mitigate error propagation over time.

Another notable characteristic, visible in Figure 5, is that the ZC model exhibits a highly symmetrical distribution for the $h_E + h_W$ CNOPs at a 9-month lead time, as well as for the NINO3 CNOPs at a 3-month lead time. In contrast, the RC model shows a more biased distribution for the estimated NINO3 CNOPs, particularly in the subcritical regime and at shorter lead times (3 months) in the supercritical regime. At longer lead times (9 months) in the supercritical regime, the RC model still shows a relatively symmetrical distribution for the estimated NINO3 CNOPs.

To better understand the origin of these differences, we classify the initial conditions analyzed (before applying the CNOPs), spanning the months [March, April, May] into three groups based on the initial eastern Pacific sea surface temperature anomalies (SSTA). Specifically, in the supercritical (subcritical) regime, we define an initial state as *positive* if the corresponding initial NINO3 index is larger than 0.2 (0.1)°C, *negative* if the initial NINO3 index is smaller than −0.2 (−0.1)°C, and *neutral* if −0.2 (−0.1)°C ≤ NINO3 ≤ 0.2 (0.1)°C. The initial states have been classified in a different way for the supercritical and subcritical regime to account for the fact in the subcritical regime the amplitudes of NINO3 is a factor 2 smaller compared to the supercritical regime see section 3.1.

As shown in Figures D2c and D2d in Appendix D, when the ZC model's SSTA initial condition in the eastern Pacific is characterized by a positive (negative) anomaly, the optimal initial perturbation for $h_E + h_W$ for a 9-month optimization period is negative (positive), if the model is initialized before the event's peak. This limits the propagation of the anomaly, leading to a weaker El Niño (La Niña) event if the model is initialized prior to the event's peak. Conversely, if the model is initialized after the event's peak, the negative (positive) thermocline perturbation drives a faster and steeper return to neutral conditions.

When the ZC model is initialized from neutral conditions, the sign of the estimated CNOPs for $h_E + h_W$ depends on the reference trajectory. If the system would have transitioned into an El Niño event, the optimal $h_E + h_W$ is negative, thus dampening the positive anomaly; if it would have transitioned into a La Niña event, the optimal $h_E + h_W$ is positive, suppressing the negative anomaly.

The same reasoning applies to the 3-month lead-time NINO3 optimal perturbations: a negative (positive) NINO3 perturbation weakens the near-term growth of an initially positive (negative) eastern Pacific SSTA.

Regarding the RC model, as shown in Figures D3c and D3d, the estimated CNOPs for the NINO3 index at a 9-month lead time in the supercritical regime exhibit a fairly symmetrical distribution overall. However, significant differences emerge

depending on whether the variable $\tau_C$ is included during training, as well as in comparison with the ZC model. When $\tau_C$ is included, and the RC is initialized before the peak of an event, the optimal NINO3 perturbation is positive (negative) for initial

conditions characterized by positive (negative) eastern Pacific SSTA, thereby reinforcing the initial anomaly and leading to larger El Niño (La Niña) events. In contrast, when the RC is initialized after the peak of a positive (negative) event, the optimal NINO3 perturbation reverses sign, resulting in a faster and steeper return to neutral conditions.

Neutral initial conditions tend to prefer negative NINO3 perturbations, resulting in stronger La Niña and weaker El Niño events relative to the reference trajectories. When $\tau_C$ is not included during training, positive SSTA initial conditions still

favor positive optimal NINO3 perturbations, yielding stronger El Niño events when the model is initialized before the peak. In contrast, for events characterized by negative initial eastern Pacific SSTA, the optimal NINO3 perturbations are positive, thereby mitigating the development of the initial negative anomaly over time. Moreover, neutral initial conditions exhibit a stronger tendency toward positive optimal NINO3 perturbations in the absence of $\tau_C$.

In the subcritical regime, the optimal NINO3 perturbation at a 9-month lead time exhibits a marked asymmetry. As illustrated

in Fig. D3a and D3b, including $\tau_C$ during training leads to consistently negative optimal perturbations across all three categories of initial conditions, whereas excluding $\tau_C$ results in positive optimal perturbations for every category. At shorter lead times (3 months), in both the subcritical and supercritical regimes and regardless of whether $\tau_C$ is included during training, the RC model clearly prefers negative optimal NINO3 perturbations. These results highlight the influence of $\tau_C$ on the estimated CNOPs, particularly in the subcritical regime. Further evidence of $\tau_C$'s impact in both supercritical and subcritical regimes

appears when examining another result shown in Fig. 5, where we find that excluding zonal surface wind-speed anomalies during training in the subcritical regime causes the RC model to exhibit greater sensitivity to thermocline depth perturbations at longer lead times compared to the supercritical regime and to the subcritical regime when $\tau_C$ is included during training. However, even in this scenario, the RC remains less sensitive to thermocline depth anomalies than the ZC model. In the subcritical regime, ENSO variability is primarily driven by atmospheric noise, introduced as stochastic wind stress forcing. This noise

affects the thermocline slope, activating mechanisms that lead to the development of perturbations. When the variable $\tau_C$ is included during training, the RC explicitly learns the relationship between wind-stress anomalies and thermocline adjustments, and the state of the surface winds is provided as an initial condition. Consequently, smaller thermocline perturbations can be amplified by wind-stress anomalies, resulting in larger deviations from the unperturbed trajectory. As a result, the optimal initial perturbations primarily target the NINO3 index, favoring negative NINO3 perturbations, while the optimal $h_E + h_W$

perturbations are only slightly negative and closer to zero, following the same trend as the NINO3 CNOPs. The impact of these optimal perturbations varies based on the initial conditions. When the RC is initialized with positive or slightly positive (neutral) eastern Pacific SSTA, negative NINO3 and $h_E + h_W$ perturbations suppress or weaken the evolution of positive initial anomalies, resulting in a faster return to neutral conditions. Conversely, when the RC is initialized with negative or slightly negative (neutral) eastern Pacific SSTA, negative NINO3 perturbations amplify the evolution of negative anomalies, leading to

either a stronger La Niña event or a longer persistence of La Niña conditions if the RC is initialized after the peak.

When $\tau_C$ is not included, the RC learns only the direct relationship between NINO3 and thermocline depth anomalies, without explicit knowledge of how wind anomalies influence thermocline slope adjustments. As a result, in the absence of

wind-forcing information, larger initial thermocline perturbations are required to induce significant error growth over time. Under these conditions, the optimal initial perturbations consist of both positive NINO3 and positive $h_E + h_W$ perturbations.

The effect of these optimal perturbations again depends on the initial conditions. When the RC is initialized with positive or slightly positive eastern Pacific SSTA, the combination of positive NINO3 and $h_E + h_W$ perturbations significantly enhances the evolution of positive anomalies, leading to a stronger El Niño event if the RC is initialized before the peak. If the RC is initialized after the peak, these perturbations extend the duration of El Niño conditions by delaying the decay of warm anomalies.

For negative or slightly negative (neutral) initial conditions, the positive optimal perturbations effectively suppress the further development of negative anomalies, once again accelerating the return to neutral conditions.

In the supercritical regime, thermocline depth perturbations are not purely activated by atmospheric noise but also emerge due to the internal instability of the system. The influence of stochastic atmospheric noise is weaker than in the subcritical case. This means that even if the model does not explicitly account for the effect of wind-stress forcing on the thermocline slope, strong initial thermocline depth anomalies are not needed to maximize error propagation. However, even in the supercritical regime, when $\tau_C$ is not included, the RC remains more sensitive to initial thermocline perturbations at longer lead times than when $\tau_C$ is included, but the difference is much less pronounced than in the subcritical regime. Also the differences in terms of the distribution of the optimal perturbations per initial conditions category is less pronounced compared to the subcritical regime and is only evident for the negative and neutral initial conditions (not for the positive ones).

## 4 Summary and Discussion

Relatively limited research has been carried out to understand the underlying reasons for the strong performance of ML prediction models in ENSO prediction, in particular their apparent ability to reduce error propagation and overcome the Spring Predictability Barrier (SPB) as deduced from dynamical models. In previous studies, explainable AI techniques like Layerwise Relevance Propagation (LRP) have been used to identify and estimate which patterns in the data are exploited by Machine Learning (ML) methods to make specific ENSO predictions (Ham et al., 2019b; Rivera Tello et al., 2023), or to explore teleconnections of ENSO (Ito et al., 2021; Liu et al., 2023b). The LRP technique has also been extended to the Echo State Network (ESN) framework to investigate the importance of the leaking rate parameter $\alpha$ and the ESN's robustness to random input perturbations while performing a El Niño/La Niña binary classification task (Landt-Hayen et al., 2022). In a recent study ( Qin et al. (2024) ), an approach similar to ours is followed, using the CNOP framework to estimate optimal initial perturbations for a U-net deep learning model trained on both reanalysis data and simulations from various CMIP6 Global Circulation Models (GCMs). They validated their results with the GFDL CM2p1 numerical model, showing that the deep learning models and numerical models exhibit a similar error evolution over time for the same initial conditions and superimposed perturbations. However, their automatic differentiation–based optimization algorithm only applies to deep learning architectures, preventing them from determining whether the deep learning and numerical models show the same optimal initial perturbations, leading to the largest error propagation.

In our study, we address this limitation by employing a gradient-free optimization algorithm (Cobyla), enabling a fair comparison between a Reservoir Computing (RC) model and the Zebiak and Cane (ZC) numerical model. Our results indicate that the RC model effectively reduces error growth from optimal initial perturbations compared to the ZC model, offering a plausible explanation for its higher predictive skill. It is important to clarify that the main objective of our study is to demonstrate that the RC can mitigate error propagation resulting from initial conditions perturbations more effectively than a classical dynamical numerical model. Such an analysis and comparison is impossible using real-world observations as we do not know the evolution operator of the real-world system and hence cannot determine the CNOP. Nevertheless, the CNOP framework can still be applied to an RC model trained on actual observations. This approach can be used to precisely assess the potential of the RC model, as well as other machine learning methodologies, to predict the real ENSO system, and to estimate their corresponding predictability limits more accurately.

Furthermore, assessing whether an RC trained with real observations exhibits similar sensitivity to specific variables as an RC trained on synthetic data from the ZC or other dynamical numerical models can provide helpful information to modelers. In particular, identifying differences in the variables to which the skill of the RC model is most sensitive can help determine whether key physical processes are being captured realistically, offering guidance on refining ENSO representation. Applying the CNOP approach to machine learning models trained on real observations could offer further benefits. For instance, while our study uses an RC model with a highly reduced input state vector consisting of just four indices, employing a more complex architecture, such as a convolutional neural network (CNN) capable of analyzing two-dimensional input fields, would allow the CNOP framework to identify the regions and variables to which the skills of the model is most sensitive. These insights could inform more precise and targeted data acquisition strategies. While such experiments with real observations are beyond the scope of the present study, they present a promising direction for future research.

In our study, we first demonstrated that the RC, when trained on data from the stochastic ZC model (acting as synthetic observations), exhibits good predictive skill up to an 18-month lead time and hence effectively overcomes the SPB problem both in the subcritical and supercritical regimes. In the supercritical regime, the RC model performs better when zonal surface wind speed anomalies are included during training while in the subcritical regime the RC actually performs better for longer lead times (9 to 18 months) when the zonal surface wind speed anomalies are excluded. While this result may depend on the implementation of the wind-stress noise (Feng and Dijkstra, 2017) which we restricted here mostly to the eastern Pacific (by using only the first EOF of the residual wind-stress field), the reason is that the RC is overfitting the noise in the subcritical regime.

Previous studies have also noticed strong predictive performances when applying the RC to ENSO forecasting. For instance, Hassanibesheli et al. (2022) achieved high prediction skills (ACC > 0.8) up to a lead time of 14 months when training the RC with the observed NINO3 and NINO3.4 indexes, decomposed into a low-frequency and high-frequency components. Their performance is comparable to ours at long lead times but our model performs better at shorter lead times (3-6 months). Additionally, like in our study, they found that their approach could mitigate the SPB problem. However, care must be taken when comparing our findings with their results due to substantial differences in the data used for training, the training variables considered, and the implementation of the forecasting framework.

After the RC's performance analysis, we investigated the propagation of errors in initial conditions in boreal spring (just before the SPB) for both the RC and deterministic ZC models using the Conditional Nonlinear Optimal Perturbation (CNOP) approach (Duan et al., 2013b). In the supercritical regime, the RC can significantly reduce error propagation in particular at longer lead times (6-9 months). In the subcritical regime, the RC is less susceptible to perturbations compared to the ZC model, when surface wind-speed anomalies are excluded during training and more susceptible when they are included. The reduced sensitivity of AI models to small initial perturbations has also been found in Selz and Craig (2023). In that study, the inability of the AI frameworks to reproduce the "butterfly effect" of the atmosphere was considered a limitation, as it prevents the generation of large ensembles due to inadequate error growth properties. However, as noted in Selz and Craig (2023) , this limitation can be mitigated by training multiple models with different random seeds to generate a confidence interval. We argue that this reduced sensitivity to initial perturbations is not a disadvantage but is what enables AI models to extend the predictability horizon of a system, allowing them to maintain higher predictive skills at longer lead times. The actual CNOPs have quite a different pattern for the ZC and RC cases; the CNOP pattern of the ZC resembles the one obtained in earlier papers (Duan et al., 2013b) with a dominant response in the thermocline field for longer lead times, but in the CNOP pattern of the RC also a strong sea surface temperature component is present.

The thermocline anomalies are important for error propagation on the longer time scales, in particular in the ZC model in which the ENSO variability is highly affected by the thermocline feedback (Zebiak and Cane, 1987). Hence, effectively, the RC model reduces the components in the thermocline anomalies and hence reduces error propagation. While we restricted to only particular cases, as we only used one value of parameter in the constraint condition $\delta$ and we allowed only one EOF in the ZC wind-stress noise, we think that the modification of the dynamical behavior in the RC (with respect to the ZC) to change the spatio-temporal properties of the error propagation is the key explanation for the superior skill of the RC on long lead times and the reason for being able to overcome the SPB. As a final remark, we specify that this mechanism is proposed to explain the RC's superior performance within a relatively short-term prediction horizon, as opposed to the decadal timescales required to assess the long-term dynamics and statistics of ENSO. Developing an emulator that captures the long-term dynamics and statistics of a system is an entirely different task, necessitating distinct model architectures, hyperparameter configurations, and evaluation criteria compared to those adopted in our study, like assessing how effectively a model captures the intrinsic nonlinearities of the system. In recent years, various Machine Learning models have demonstrated impressive predictive skills up to a 21-month lead time without necessarily capturing all the underlying physical processes of ENSO. The results shown here for the Reservoir Computer can be extended to other Machine Learning models, potentially explaining their predictive skills up to nearly two years lead time, far beyond the SPB.

*Code availability.* All data and code used in this study are available at this link https://zenodo.org/records/15006826 (Guardamagna (2025))

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

## Appendix A: Zebiak and Cane model

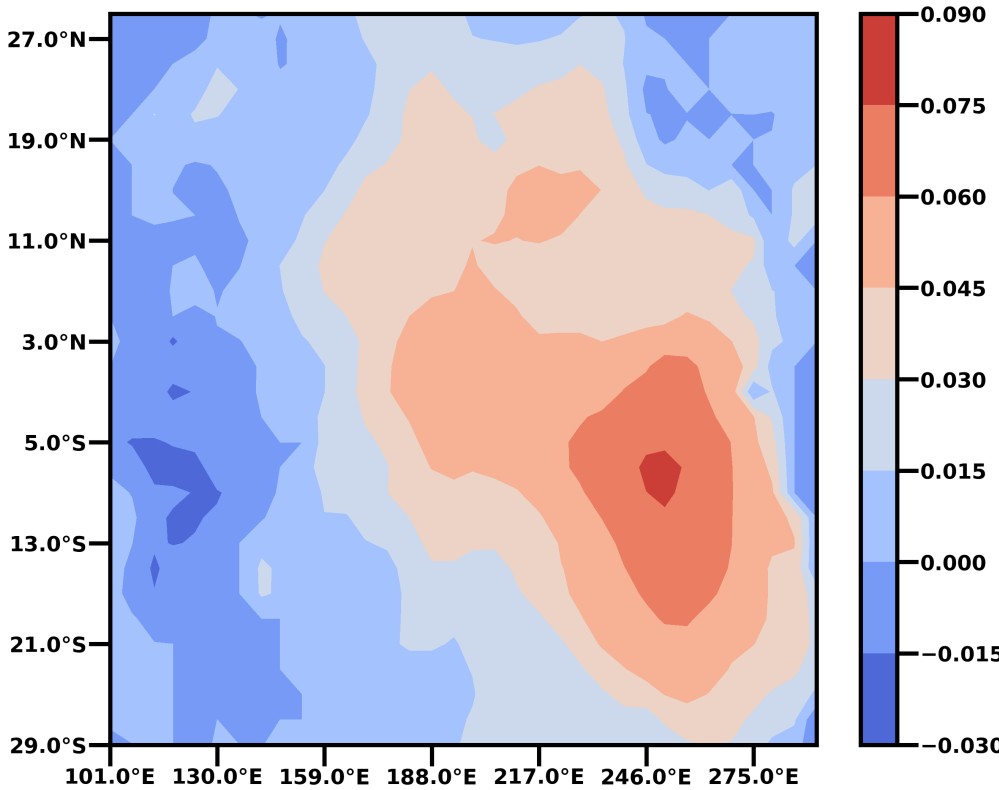

**Figure A1.** First EOF of the residual zonal wind stress anomalies as determined from the ORAS5 dataset (Copernicus Climate Change Service (2021)).

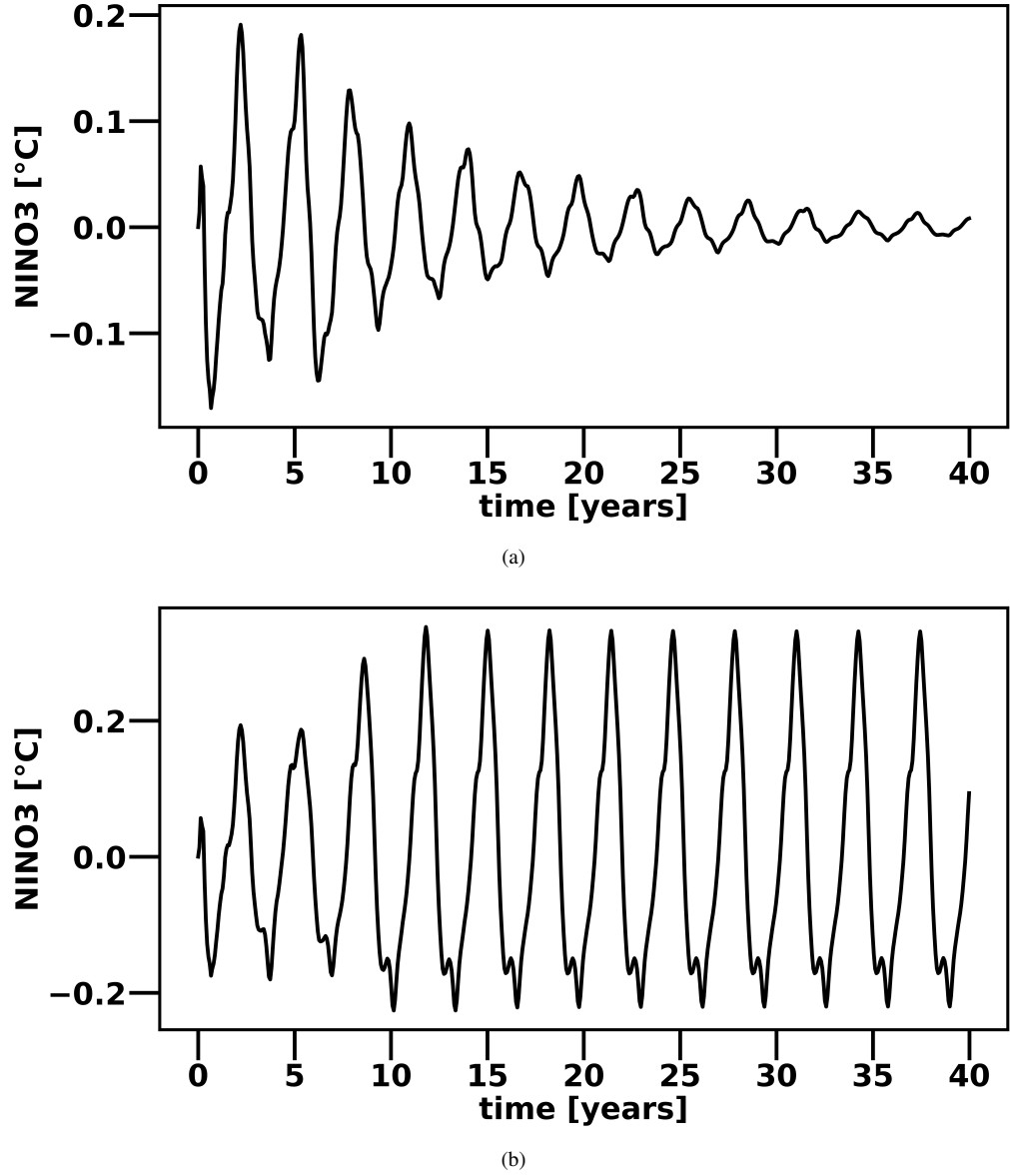

**Figure A2.** NINO3 index from the deterministic ZC model for (a) $r_d = 0.79$ and (b) $r_d = 0.8$.

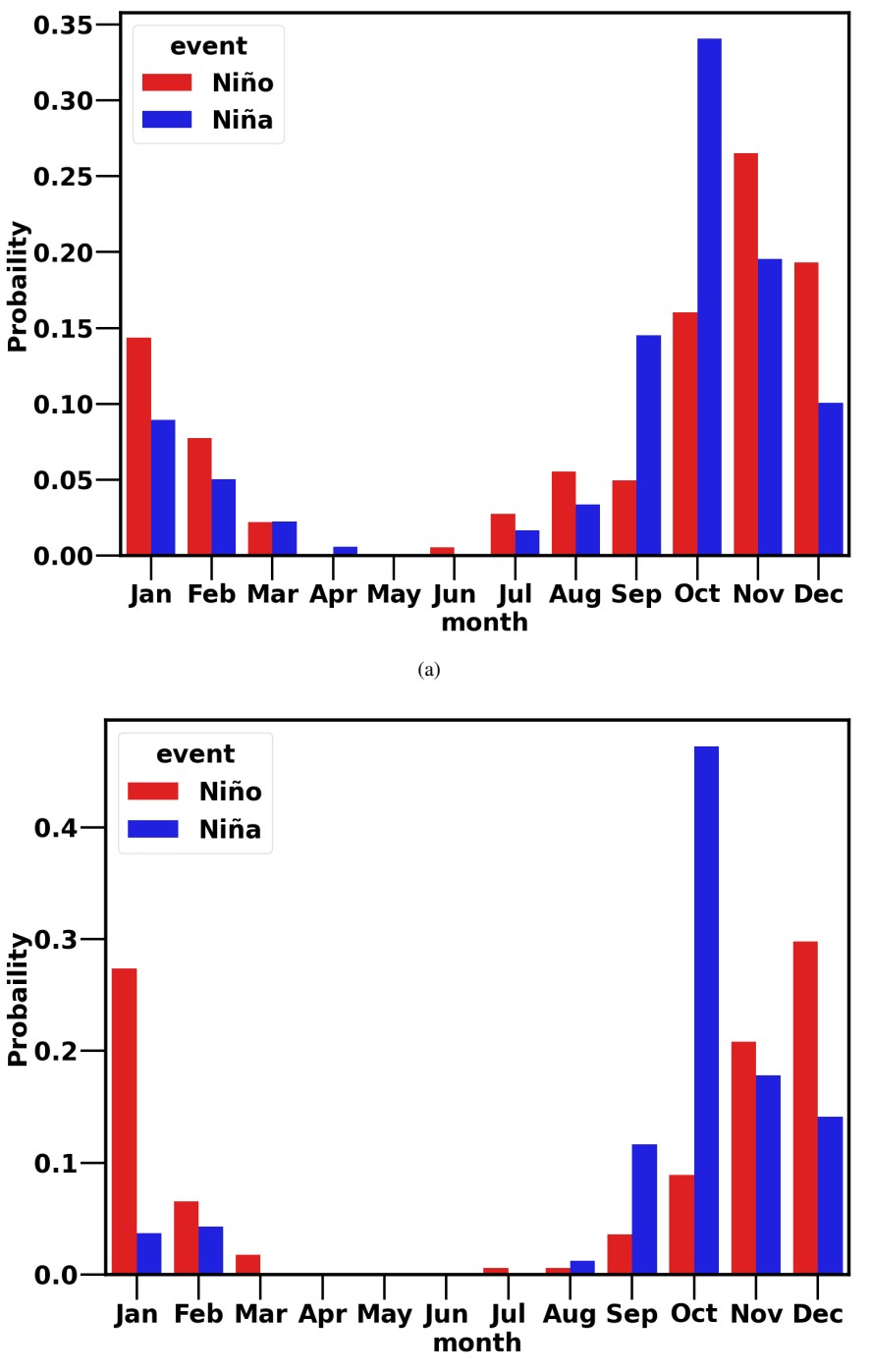

**Figure A3.** Frequency of the occurrence of La Niña and El Niño events for each calendar month. (a) $r_d = 0.77$ (b) $r_d = 0.9$. For both $r_d$ values a stochastic ZC model realization of 1000 years has been considered.

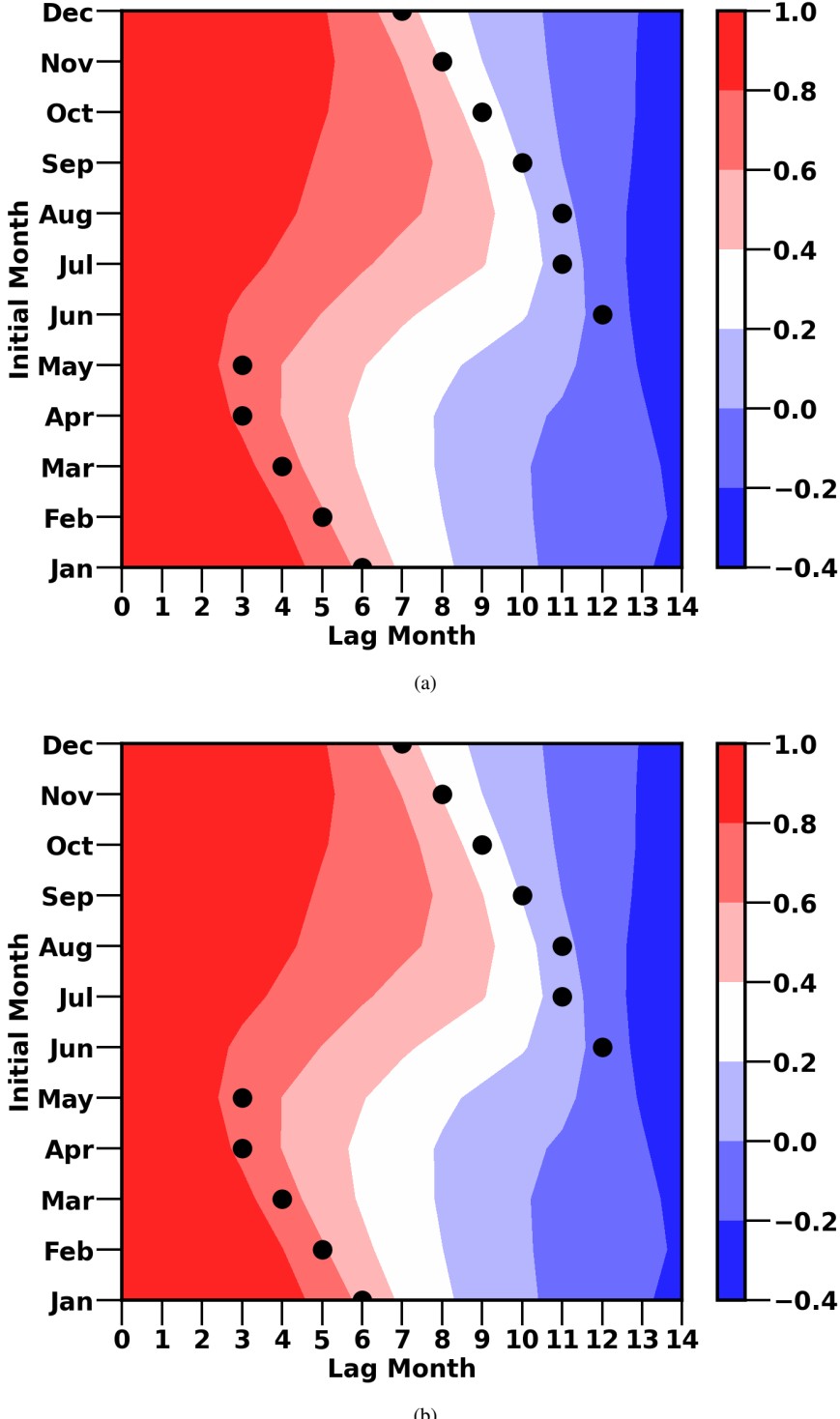

**Figure A4.** Level of autocorrelation of the NINO3 index for different starting months for (a) $r_d = 0.77$ and (b) $r_d = 0.9$. The black dots indicate the lag month corresponding to the maximum decrease in autocorrelation. In each case, a 1000 years stochastic ZC model realization, has been considered

## Appendix B: COBYLA Algorithm

The Constrained Optimization BY Linear Approximations (COBYLA) algorithm (Powell (1994)) is a gradient-free optimization method designed to solve nonlinear constrained optimization problems. Given an objective function $J(x)$ to be minimized, where $x \in \mathbb{R}^n$, and a set of nonlinear constraints $C_i(x) \leq 0$, the algorithm starts from an initial guess $x_0$ and constructs an initial $n$-dimensional simplex, represented by a set of $n+1$ vertices $V^{(0)} = \{v_1^{(0)}, v_2^{(0)}, ..., v_{n+1}^{(0)}\} \subset \mathbb{R}^n$. Each vertex is defined as $v_i^{(0)} = x_0 + \rho_{\text{beg}} e_i$, where $e_i$ is the $i$-th coordinate vector (a standard basis vector in $\mathbb{R}^n$), and $\rho_{\text{beg}}$ is a specified initial trust-region radius that determines the initial simplex size.

At each iteration, the algorithm constructs a linear approximation of both the objective function and the constraints using a linear interpolation at the $n+1$ current simplex vertices $V^{(i)}$. It then identifies the worst-performing vertex, denoted as $v_{\text{worst}}^{(i)} = \arg\max_{v \in V^{(i)}} J(v)$. Once the worst-performing vertex is found, the algorithm formulates and solves a linear optimization problem within a trust region of radius $\rho$ around $v_{\text{worst}}^{(i)}$ to generate a new candidate vertex $v_{\text{new}}^{(i+1)}$. If this new vertex improves the objective function, it replaces $v_{\text{worst}}^{(i)}$, updating the simplex structure. Otherwise, the trust-region radius is reduced, and the linear optimization problem is solved again.

The algorithm continues iterating until one of the stopping criteria is met. It terminates when the trust-region radius $\rho$ falls below a predefined threshold $\rho_{\text{end}}$, when the change in the objective function is smaller than a specified tolerance $\varepsilon$, or when the number of iterations reaches the maximum allowed value $N_{\text{max}}$. COBYLA is particularly well-suited for nonlinear optimization problems with a small number of variables, especially when computing derivatives is challenging or infeasible. These characteristics make COBYLA an ideal choice for our analysis, as the ZC model's derivatives are impossible to compute, and our optimization problem involves only three variables (see Sections 2.3 and 3.3).

To validate the estimated CNOPs obtained using COBYLA for both the RC and ZC models, we first confirm that the CNOPs lie on the boundary of the sphere defined by the constraints. Next, we evaluate error propagation by applying multiple randomly chosen initial perturbations sampled from this boundary, to determine whether the CNOPs indeed correspond to the largest error growth. Figures B1 and B2 present these validation results. Specifically, Figure B1 shows a scatter plot of all estimated CNOPs for both models in the three-dimensional space defined by the normalized NINO3, $h_E$ and $h_W$ optimal initial perturbations. In contrast, Figure B2 illustrates, for a representative case, the divergence between perturbed and unperturbed trajectories resulting from both the CNOP and 50 random initial perturbations sampled along the constraint boundary. For illustration, results are provided for five different years, with both the RC and ZC models always initialized in April.

For our implementation, we adopted the COBYLA solver from the SciPy Python library (Virtanen et al. (2020)). Since COBYLA is inherently designed to solve minimization problems, while our objective is to maximize the distance between the reference and perturbed trajectories (see Sections 2.3 and 3.3), we account for this by minimizing $-J(x)$ instead of $J(x)$.

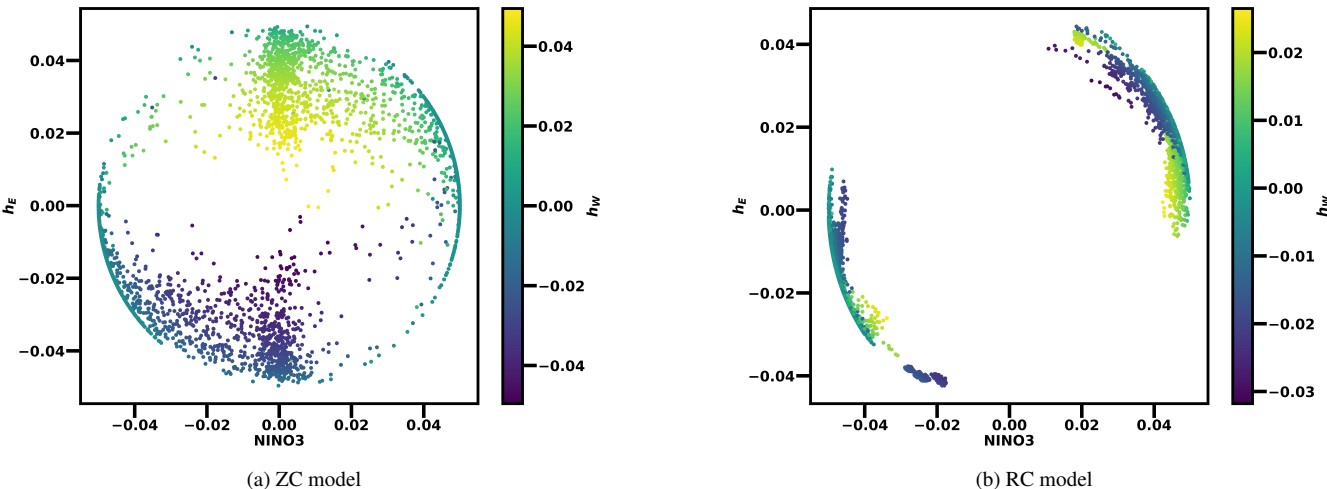

(a) ZC model               (b) RC model

**Figure B1.** Scatter plot of all computed CNOPs in the normalized NINO3 index vs. [hE, hW] anomaly plane for (a) the ZC model and (b) the RC model for both $\tau_C$ included and excluded during training. The NINO3 index is normalized by 2°C, while the [hE, hW] anomalies are normalized by 50m.

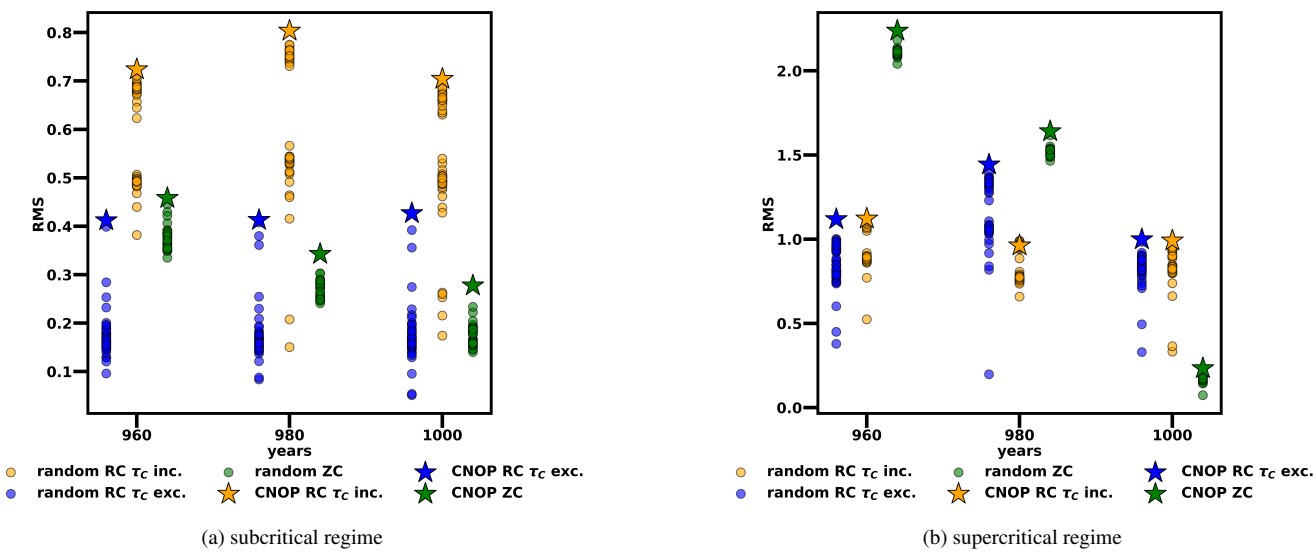

(a) subcritical regime          (b) supercritical regime

**Figure B2.** Scatter plot of the RMS differences between perturbed and unperturbed trajectories for three model years, with both the RC and ZC models always initialized in April. Results are presented for the CNOP (i.e., the optimal initial perturbation) and for 50 random initial perturbations sampled from the boundary of the constraints.

| Input Variables | $r_d$ | Best Hyperparameters |
|---|---|---|
| **NINO3, $h_E$, $h_W$, $\tau_C$** | **0.9** | $N_x = 391$, $\rho = 0.84$, $< k >= 0.16$, $a = 0.8$, $\alpha = 0.57$ |
| **NINO3, $h_E$, $h_W$** | **0.9** | $N_x = 404$, $\rho = 1.07$, $< k >= 0.2$, $a = 0.58$, $\alpha = 0.39$ |
| **NINO3, $h_E$, $h_W$, $\tau_C$** | **0.77** | $N_x = 476$, $\rho = 0.8$, $< k >= 0.1$, $a = 0.63$, $\alpha = 0.6$ |
| **NINO3, $h_E$, $h_W$** | **0.77** | $N_x = 534$, $\rho = 0.88$, $< k >= 0.1$, $a = 0.38$, $\alpha = 0.17$ |

**Table C1.** Table showing the optimal RC model's hyperparameters sets, for the supercritical ($r_d = 0.9$) and subcritical ($r_d = 0.77$) regimes, and each input variable configuration (with and without the inclusion of $\tau_C$).

**Appendix D:  CNOPs**

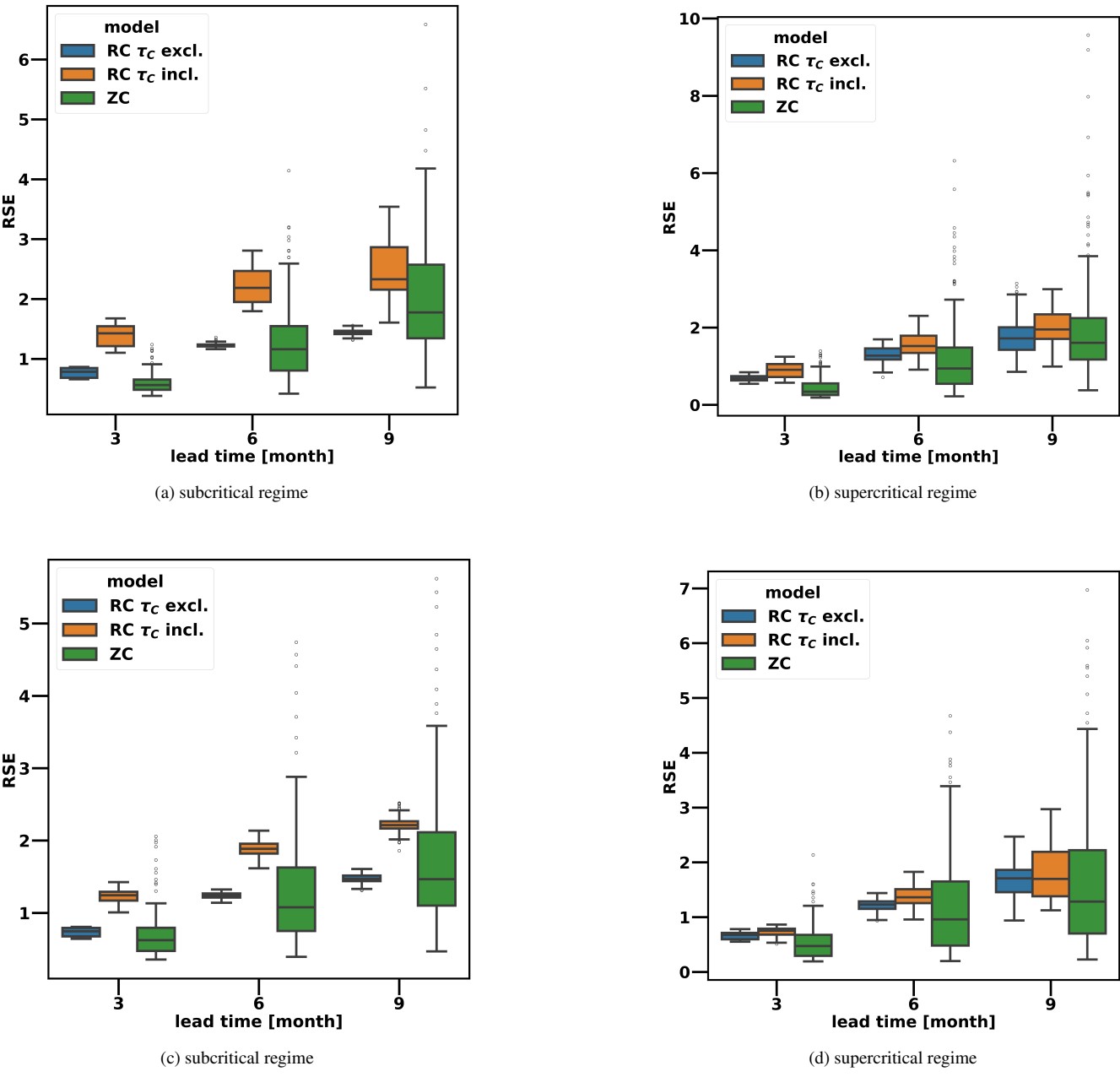

(a) subcritical regime

(b) supercritical regime

(c) subcritical regime

(d) supercritical regime

**Figure D1.** Distribution of the normalized RSE distances between perturbed and unperturbed trajectories for different lead times with the application of CNOPs, using initial conditions from: (a)-(b) [December,January,February], and (c)-(d) [September,October,November]. The left plots (a) and (c) display results for $r_d = 0.77$, while the right plots (b) and (d) show results for $r_d = 0.9$. The boxes indicate the interquartile range (IQR), the range within the central 50% of data points lie. The whiskers extend to the minimum and maximum values within 1.5 times the IQR from the first and third quartile. The central line corresponds to the median. The RSE distances are normalized by the standard deviation of the NINO3 index extracted from the 50 years of synthetic observations considered for CNOPs computation (0.29 for the subcritical regime and 0.56 for the supercritical regime).

## Subcritical Regime ($r_d = 0.77$)

| Start/Lead | 3 Months | 6 Months | 9 Months |
|:---:|:---:|:---:|:---:|
| Spring | 0.6 | 1.52 | 1.88 |
| Winter | 0.56 | 1.16 | 1.78 |
| Autumn | 0.62 | 1.08 | 1.47 |

## Supercritical Regime ($r_d = 0.9$)

| Start/Lead | 3 Months | 6 Months | 9 Months |
|:---:|:---:|:---:|:---:|
| Spring | 0.41 | 1.41 | 2.07 |
| Winter | 0.34 | 0.94 | 1.6 |
| Autumn | 0.48 | 0.96 | 1.28 |

**Blue**: Forecast crosses the SPB   **Orange**: Forecast does not cross the SPB

**Table D1.** Table showing the median of the normalized RSE distances between perturbed and unperturbed trajectories at various lead times, with CNOPs applied across different starting seasons. Only the Zebiak and Cane model is considered, with the top table representing the subcritical regime ($r_d = 0.77$) and the bottom table representing the supercritical regime ($r_d = 0.9$). All RSE distances are normalized by the standard deviation of the NINO3 index from the 50 years of synthetic observations considered for the CNOPs computation (0.29 for the subcritical regime and 0.56 for the supercritical regime).

## Subcritical Regime, $\tau_C$ Included During Training

| Model | Start/Lead | 3 Months | 6 Months | 9 Months |
|---|---|---|---|---|
| RC | Spring | 1.35 (0.14) | 2.07 (0.33) | 2.4 (0.54) |
| | Winter | 1.42 (0.33) | 2.19 (0.52) | 2.33 (0.71) |
| | Autumn | 1.24 (0.12) | 1.89 (0.14) | 2.21 (0.1) |
| ZC | Spring | 0.6 (0.28) | 1.52 (1.49) | 1.88 (1.86) |
| | Winter | 0.56 (0.17) | 1.16 (0.74) | 1.78 (1.23) |
| | Autumn | 0.62 (0.32) | 1.08 (0.88) | 1.47 (1.01) |

## Subcritical Regime, $\tau_C$ Excluded During Training

| Model | Start/Lead | 3 Months | 6 Months | 9 Months |
|---|---|---|---|---|
| RC | Spring | 0.76 (0.11) | 1.2 (0.05) | 1.44 (0.08) |
| | Winter | 0.79 (0.16) | 1.22 (0.03) | 1.44 (0.06) |
| | Autumn | 0.75 (0.12) | 1.24 (0.05) | 1.47 (0.08) |
| ZC | Spring | 0.6 (0.28) | 1.52 (1.49) | 1.88 (1.86) |
| | Winter | 0.56 (0.17) | 1.16 (0.74) | 1.78 (1.23) |
| | Autumn | 0.62 (0.32) | 1.08 (0.88) | 1.47 (1.01) |

**Blue**: Forecast crosses the SPB    **Orange**: Forecast does not cross the SPB

## Supercritical Regime, $\tau_C$ Included During Training

| Model | Start/Lead | 3 Months | 6 Months | 9 Months |
|-------|-----------|----------|----------|----------|
| RC | Spring | 0.86 (0.09) | 1.46 (0.22) | 1.82 (0.32) |
| RC | Winter | 0.91 (0.33) | 1.52 (0.44) | 1.95 (0.64) |
| RC | Autumn | 0.75 (0.11) | 1.36 (0.25) | 1.7 (0.8) |
| ZC | Spring | 0.41 (0.28) | 1.41 (1.36) | 2.07 (1.85) |
| ZC | Winter | 0.34 (0.3) | 0.94 (0.94) | 1.6 (1.07) |
| ZC | Autumn | 0.48 (0.38) | 0.96 (1.17) | 1.28 (1.52) |

## Supercritical Regime, $\tau_C$ Excluded During Training

| Model | Start/Lead | 3 Months | 6 Months | 9 Months |
|-------|-----------|----------|----------|----------|
| RC | Spring | 0.67 (0.07) | 1.3 (0.23) | 1.65 (0.52) |
| RC | Winter | 0.68 (0.11) | 1.27 (0.29) | 1.72 (0.58) |
| RC | Autumn | 0.68 (0.12) | 1.23 (0.14) | 1.7 (0.41) |
| ZC | Spring | 0.41 (0.28) | 1.41 (1.36) | 2.07 (1.85) |
| ZC | Winter | 0.34 (0.3) | 0.94 (0.94) | 1.6 (1.07) |
| ZC | Autumn | 0.48 (0.38) | 0.96 (1.17) | 1.28 (1.52) |

**Blue**: Forecast crosses the SPB   **Orange**: Forecast does not cross the SPB

**Table D2.** Median (IQR) of the normalized RSE distances between perturbed and unperturbed trajectories at various lead times, with CNOPs applied across different starting seasons. Both ZC and RC models (trained with and without $\tau_C$) are considered in the subcritical ($r_d = 0.77$) and supercritical ($r_d = 0.9$) regime. RSE distances are normalized by the standard deviation of NINO3 index from the 50 years of synthetic observations considered for CNOPs computation (0.29 for the subcritical regime and 0.56 for the supercritical regime). The interquartile range IQR is defined as the distance between the first and third quartile.

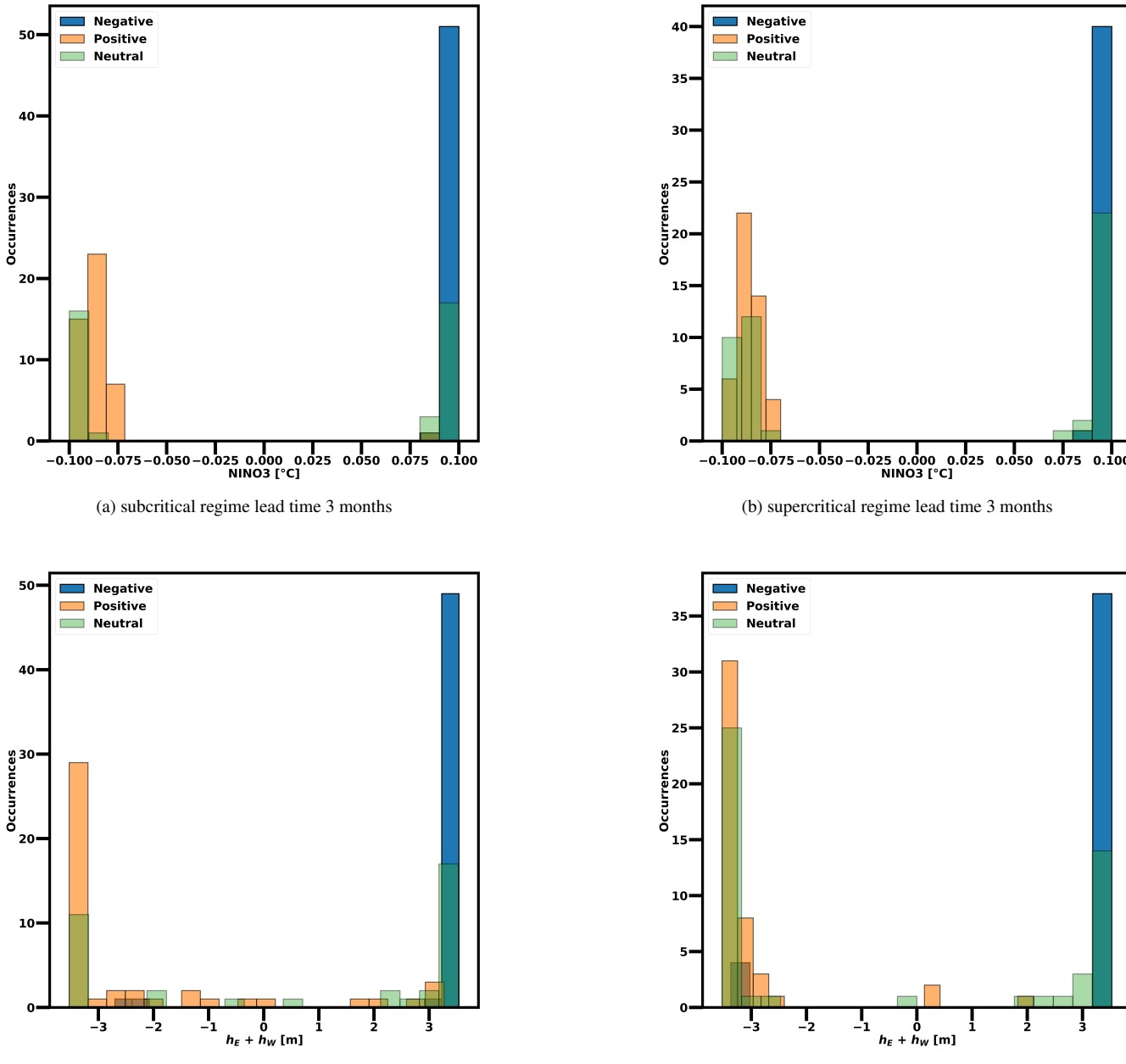

**Figure D2.** Distribution of the ZC model's CNOPS for positive, negative, and neutral eastern Pacific SSTA initial conditions. Panels (a)-(b) (top row) show the CNOPs obtained for the NINO3 index at a 3-month lead time, comparing subcritical and supercritical regimes, while panels (c)-(d) (bottom row) show the CNOPs obtained for $h_E + h_W$, the sum of the thermocline anomalies in the eastern and western Pacific at a lead time of 9 months. In the supercritical regime, positive, negative, and neutral eastern Pacific SSTA initial conditions are defined as NINO3 $\geq 0.2$, NINO3 $\leq -0.2$, and $-0.2 <$ NINO3 $< 0.2$, respectively. In the subcritical regime, positive, negative, and neutral SSTA initial conditions are defined as NINO3 $\geq 0.1$, NINO3 $\leq -0.1$, and $-0.1 \leq$ NINO3 $< 0.1$, respectively. In every case the months [March,April,May] (Spring Season) from the last 50 years of the synthetic data used for testing (see section 3.1) are taken as initial conditions, yielding a total of 150 initial conditions considered.

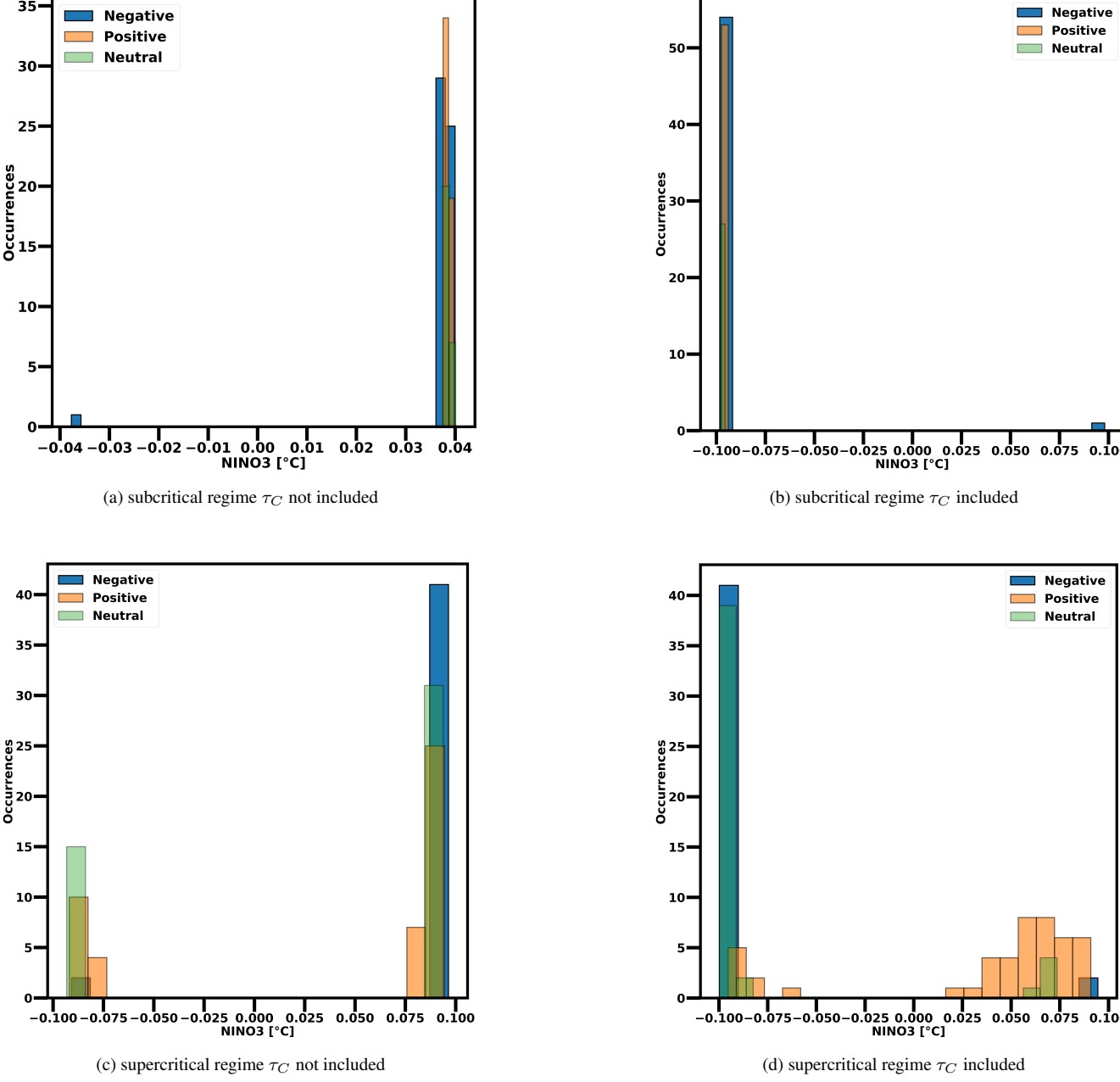

**Figure D3.** Distribution of the RC model's CNOPs for positive, negative, and neutral eastern Pacific SSTA initial conditions. Panels (a)–(b) (top row) display the CNOPs for the NINO3 index at a 9-month lead time under the subcritical regime, with and without $\tau_C$ included during training. Panels (c)–(d) (bottom row) show the CNOPs for the supercritical regime, also at a 9-month lead time, with $\tau_C$ either included or excluded. In the supercritical regime, positive, negative, and neutral initial conditions are defined as NINO3 $\geq 0.2$, NINO3 $\leq -0.2$, and $-0.2 <$ NINO3 $< 0.2$, respectively. In the subcritical regime, positive, negative, and neutral SSTA initial conditions are defined as NINO3 $\geq 0.1$, NINO3 $\leq -0.1$, and $-0.1 \leq$ NINO3 $< 0.1$, respectively. In every case the months [March,April,May] (Spring Season) from the last 50 years of the synthetic data used for testing (see section 3.1) are taken as initial conditions, yielding a total of 150 initial conditions considered.

*Author contributions.* All authors contributed to the design of this study. FG carried out all the computations and produced a first draft the paper. All authors contributed to the interpretation of the results and the final version of the paper.

*Competing interests.* The authors declare that no competing interests are present.

*Acknowledgements.* The work of Francesco Guardamagna, Claudia Wieners and Henk Dijkstra was supported by the Netherlands Organization for Scientific Research (NWO) under grant OCENW.M20.277.