# Peer review of "Explaining the high skill of Reservoir Computing methods in El Niño prediction"

_Nonlinear Processes in Geophysics, 2024_

## Referee Comment (RC1)

**Comments on npg-2024-24**

This study focuses on explaining the origins of high forecasting skills of recently emerging deep learning models for ENSO predictions. More specifically, the authors firstly build a skillful Reservoir Computers (RC) model for simulating Zebiak-Cane numerical model, and then investigate and compare the sensitivities on initial perturbations of RC and ZC models (also including an LC model). The authors find RC models are less susceptible to initial perturbations no matter for short and long lead months, which is also the possible reason of weakening the impact of spring predictability barrier (SPB) and extending the effective lead time. In general, this study crafts some novel experiments, and I have the following major questions for the author to answer:

1. $W^{in}$ in Equation (4a) and $W_{in}$ in the following illustration is not consistent.

2. What is the value of $N_x$ in your study, which is quite an important configuration for forecasting skill of RC model.

3. Around line 169, the authors mention that "the initial reservoir state for each prediction was determined using 5-year of data prior to the time $t$". Can you explain this expression more to those unfamiliar with RC models?

4. Around line 224, the authors mention that "This result aligns with expectations, as non-linearities play a more important role in the supercritical regime". I am wondering is it true that the supercritical regime exhibits more nonlinear than subcritical regime, which favors the performance of the RC model? What's the relationship between nonlinearity of regime and RC model performance?

5. Around line 248, do the authors use values of the CNOP objective function to assess whether the model is susceptible?

6. For figure 5, why the CNOP results for ZC model is quite symmetrical while the CNOP results for RC models are usually biased?

7. Around line 297, the authors mention that the inclusion or exclusion of wind speed anomalies has a large effect on the different variables of CNOP in the RC models for the subcritical regime. I am wondering why this is not obvious in the RC models for the supercritical regime?

8. I have noticed there is another similar study that revealing the initial perturbations of SST for ENSO predictions in AI model (https://doi.org/10.1002/qj.4882). Maybe this is a more comprehensive way to detecting the detailed patterns and physical variables of initial perturbations related to SPB from index models (such as RC models used in this study) to spatial models.

9. The AI model appears to be less sensitive to the initial perturbations, or the initial perturbations do not grow as fast as those in numerical models, which is the reason for the higher skill of the AI model. Similar conclusions are also obtained in another similar study (https://doi.org/10.1029/2023GL105747). Can the authors discuss the pros and cons of this characteristics? I think this will be significantly valuable for the future modelling, as well as further understanding, of earth system.

---

## Author Comment (AC1)

**MS-No.:npg-2024-24**

**Title:** Explaining the high skill of Reservoir Computing method in El Niño prediction

**Authors:** Francesco Guardamagna, Claudia E. Wieners and Henk A. Dijkstra

**Point-by-point reply to reviewer #1**

February 9, 2025

We thank the reviewer for their careful reading and for the useful comments on the manuscript.

**Overview**

*This study focuses on explaining the origins of high forecasting skills of recently emerging deep learning models for ENSO predictions. More specifically, the authors firstly build a skillful Reservoir Computers (RC) model for simulating Zebiak-Cane numerical model, and then investigate and compare the sensitivities on initial perturbations of RC and ZC models (also including an LC model). The authors find RC models are less susceptible to initial perturbations no matter for short and long lead months, which is also the possible reason of weakening the impact of spring predictability barrier (SPB) and extending the effective lead time. In general, this study crafts some novel experiments, and I have the following major questions for the author to answer:*

**Major comments:**

1. $W_{in}$ *in Equation (4a) and* $W^{in}$ *in the following illustration is not consistent.*

   **Author's reply:**
   We thank the reviewer for pointing out this imprecision.

   **Changes in manuscript:**
   We will correct this imprecision in lines 99, 100, and 101 on page 4 of the revised manuscript.

2. *What is the value of $N_x$ in your study, which is quite an important configuration for forecasting skill of RC model.*

   **Author's reply:**

   For both the supercritical and subcritical regimes, as well as for each set of training variables, we determined the optimal hyperparameter sets using a Bayesian search. The search consistently converged on large reservoir dimensions $N_x$, with notable differences between the supercritical and subcritical regimes. In the supercritical regime, the optimal reservoir dimension is approximately 400, regardless of whether zonal surface wind speed anomalies ($\tau_C$) are included in the training variable set. In the subcritical regime, the optimal reservoir dimension is larger, with $N_x = 476$ when $\tau_C$ is included and $N_x = 534$ when $\tau_C$ is excluded.

   **Changes in manuscript:**
   We will specify the $N_x$ values used in our study in the "Training and validation of the RC" section on page 8. Moreover, a summary table reporting the optimal hyperparameter sets for each regime and training variable set will be added to the Appendix.

3. *Around line 169, the authors mention that "the initial reservoir state for each prediction was determined using 5 years of data before the time t". Can you explain this expression more to those unfamiliar with RC models?*

   **Author's reply:**
   We agree with the reviewer that this expression may not be clear to those unfamiliar with the Reservoir Computer (RC) model. Discarding the initial $x(n)$ reservoir states for $0 \leq n \leq n_{transient}$ before training and evaluation is a standard practice in Reservoir Computing. This step is necessary to mitigate the impact of initial transients caused by the arbitrary initialization of the reservoir state, which is typically set to $x(0) = 0$ or initialized randomly. In our case, the reservoir state was initialized as $x(0) = 0$. This initialization creates an artificial starting state that is unlikely to recur once the reservoir dynamics stabilize. A warmup period is therefore introduced to allow the reservoir to reach a

stable dynamical regime before training or inference. The length of the warmup period depends on the reservoir's memory capacity and the specific learning task. Based on our experiments, a warmup period of 5 years is sufficient to stabilize the reservoir dynamics and eliminate the effects of initial transients. As a result, before inference, the reservoir states corresponding to the 5 years preceding the initial time step $t(n)$ (the starting point of our forecast) are discarded. In our notation, this means discarding $x(n)$ reservoir states for $t(n) - n_{transient} \leq n \leq t(n)$, where $n_{transient} = 180$, given our 10-days time step.

**Changes in manuscript:**
We will better explain why, when working with a RC model, a warm-up period is necessary to properly initialize the Reservoir state in the "Training and validation of the RC" section on page 8.

4. *Around line 224, the authors mention that "This result aligns with expectations, as non-linearities play a more important role in the supercritical regime". I am wondering is it true that the supercritical regime exhibits more nonlinear than subcritical regime, which favors the performance of the RC model? What's the relationship between nonlinearity of regime and RC model performance?*

**Author's reply:**
ENSO can be described by two different theoretical frameworks. According to one perspective, it is a stable (damped) mode sustained primarily by random atmospheric noise (subcritical regime). Alternatively, it can be viewed as a self-sustained oscillatory mode (supercritical regime). In the latter scenario, nonlinearity is essential in modulating ENSO behavior. In the Zebiak and Cane (ZC) model, nonlinearities come from three main sources: heat advection, wind stress anomalies, and subsurface water temperature variations [1].

The Reservoir Computing model can capture complex nonlinear relationships between input variables by employing a nonlinear activation function (in our study, the hyperbolic tangent). In contrast, a Linear Regressor can only estimate linear relationships between input variables. Consequently, the performance gap between the Reservoir

Computing model and the Linear Regressor is expected to be more pronounced in the supercritical regime, where nonlinear effects are more important.

**Changes in manuscript:**
We will provide a clearer explanation of why nonlinearities play a more significant role in the supercritical regime compared to the subcritical regime in the "Zebiak and Cane" model section on page 2. Additionally, in the "Reservoir Computer" section on page 4, we will explain why the Reservoir Computing model is better suited for solving problems involving nonlinear relationships between input variables, compared to a simple Linear Regressor.

5. *Around line 248, do the authors use values of the CNOP objective function to assess whether the model is susceptible?*

**Author's reply:**

For both the RC and the ZC model, we used the Cobyla optimization algorithm to maximize the CNOP objective function and estimate the optimal initial perturbations. The resulting maximal error growth, calculated for a specific lead time, was then taken as our measure of the model's sensitivity to initial perturbations.

**Changes in manuscript:**
We will better clarify this point in the "CNOPs Analysis" section on page 12.

6. *For figure 5, why the CNOP results for ZC model is quite symmetrical while the CNOP results for RC models are usually biased?*

**Author's reply:**
The ZC model's optimal initial perturbations exhibit a notably symmetrical distribution, evident in both SST perturbations at shorter lead times (3 months) and thermocline depth perturbations at longer lead times (6 and 9 months). This symmetry suggests that the model is

sensitive to both negative and positive initial perturbations, depending on the specific event.

In contrast, the RC generally demonstrates greater sensitivity to initial SST perturbations across both shorter and longer lead times, and the distribution of these optimal initial SST perturbations consistently shows a clear preference for either positive or negative values.

To better understand these differences in behavior, we plan to identify the specific initial conditions and times of year when the ZC model shows a preference for negative or positive perturbations. We will then compare the behavior of the RC for the same types of events, focusing on how the optimal perturbations evolve for both models. This additional analysis will provide valuable insights into why the ZC model's optimal initial perturbations exhibit a more symmetrical distribution.

**Changes in manuscript:**
We will add and discuss the results of this additional analysis in the "CNOP analysis" section of the revised manuscript on page 11.

7. *Around line 297, the authors mention that the inclusion or exclusion of wind speed anomalies has a large effect on the different variables of CNOP in the RC models for the subcritical regime. I am wondering why this is not obvious in the RC models for the supercritical regime?*

**Author's reply:**

In the subcritical regime, ENSO variability is primarily sustained by atmospheric noise, introduced as stochastic wind stress forcing. This noise influences the thermocline slope, activating mechanisms that lead to the development of perturbations. When the variable $\tau_c$ is included during training, the RC explicitly learns the relationship between wind anomalies and thermocline adjustments, and the state of the surface winds is provided as an initial condition. Consequently, smaller thermocline perturbations can be amplified by wind anomalies, leading to larger deviations from the reference trajectory. In contrast, when $\tau_c$ is not included, the RC only learns the direct relationship between SST and thermocline depth anomalies without explicit knowledge of how

wind anomalies influence thermocline slope adjustments. As a result, in the absence of wind-forcing information, a larger initial thermocline perturbation is required to generate significant error propagation over time.

In the supercritical regime, thermocline depth perturbations are not purely activated by atmospheric noise but also emerge due to the internal instability of the system. The influence of stochastic atmospheric noise is weaker than in the subcritical case. This means that even if the model does not explicitly account for the effect of wind forcing on the thermocline slope, strong initial thermocline depth anomalies are not needed to maximize error propagation.

However, even in the supercritical regime, when $\tau_c$ is not included, the RC remains more sensitive to initial thermocline perturbations at longer lead times than when $\tau_c$ is included, but the difference is much less pronounced than in the subcritical regime.

In every case, the Reservoir Computer is consistently less sensitive to thermocline depth perturbations at longer lead times compared to the Zebiak and Cane model. This suggests that the RC effectively mitigates error propagation from thermocline perturbations.

**Changes in manuscript:**

We will provide a clearer explanation in the "CNOPs Analysis" section on page 11 of why the inclusion of $\tau_c$ has a greater impact on the RC's optimal initial perturbations in the subcritical regime than in the supercritical regime.

8. *I have noticed there is another similar study that revealing the initial perturbations of SST for ENSO predictions in AI model (https://doi.org/10.1002/qj.4882). Maybe this is a more comprehensive way to detecting the detailed patterns and physical variables of initial perturbations related to SPB from index models (such as RC models used in this study) to spatial models.*

**Author's reply:**

We thank the reviewer for bringing to our attention a study similar to ours that was not cited in our manuscript. Although the analysis by Qin et al. [2] bears similarities to our experiments, there are substantial differences that make both studies valuable:

(a) In [2], the GFDL CM2p1 dynamical numerical model is used solely to validate the optimal initial perturbations computed for the Deep Learning model employed in their study. However, they do not compute the optimal initial perturbations for the GFDL CM2p1 model itself. As a result, their analysis does not explore whether the GFDL CM2p1 and Deep Learning models exhibit similar optimal initial perturbations or how optimal initial errors propagate in both models. This comparison is a central aspect of our study.

(b) In [2], the CNOP objective function for the deep-learning model is optimized using automatic differentiation, a feature available in most modern deep-learning frameworks, such as PyTorch and TensorFlow. However, this approach does not apply to dynamical numerical models since these models are not implemented using modern deep-learning frameworks. Moreover, for these models, computing or approximating gradients with respect to their outputs is often highly complex or practically infeasible, making it challenging to apply gradient-based optimization techniques. To enable a meaningful comparison of how optimal initial perturbations evolve in both machine learning and numerical models, a gradient-free optimization method, like the Cobyla algorithm employed in our study, is essential.

These differences in the analysis performed and the methods adopted underscore the complementary contributions of our study and that in [2].

**Changes in manuscript:**

In the "Summary and Discussion" section, we will reference [2], discuss their interesting results and underline the key differences between their

study and ours.

9. *The AI model appears to be less sensitive to the initial perturbations, or the initial perturbations do not grow as fast as those in numerical models, which is the reason for the higher skill of the AI model. Similar conclusions are also obtained in another similar study (https://doi.org/10.1029/2023GL105747). Can the authors discuss the pros and cons of this characteristics? I think this will be significantly valuable for the future modelling, as well as further understanding, of earth system.*

**Author's reply:**
We agree with the referee that discussing the pros and cons of this characteristic of AI models compared to dynamical models will be highly valuable.

**Changes in manuscript:**
In the "Summary and Discussion" section, we will discuss the pros and cons of this characteristic of AI models, referring to the interesting results presented by Selz et al. in [3].

**References**

[1] Wansuo Duan, Yanshan Yu, Hui Xu, and Peng Zhao. Behaviors of nonlinearities modulating the el niño events induced by optimal precursory disturbances. *Climate Dynamics*, 40(5):1399–1413, March 2013.

[2] Bo Qin, Zeyun Yang, Mu Mu, Yuntao Wei, Yuehan Cui, Xianghui Fang, Guokun Dai, and Shijin Yuan. The first kind of predictability problem of el niño predictions in a multivariate coupled data-driven model. *Quarterly Journal of the Royal Meteorological Society*, 150(765):5452–5471, 2024.

[3] T. Selz and G. C. Craig. Can artificial intelligence-based weather prediction models simulate the butterfly effect? *Geophysical Research Letters*, 50(20):e2023GL105747, 2023. e2023GL105747 2023GL105747.

---

## Author Comment (AC2)

**MS-No.:npg-2024-24**

**Title:** Explaining the high skill of Reservoir Computing method in El Niño prediction

**Authors:** Francesco Guardamagna, Claudia E. Wieners and Henk A. Dijkstra

**Point-by-point reply to reviewer #2**

February 9, 2025

We thank the reviewer for their careful reading and for the useful comments on the manuscript.

**Overview**

*Reservoir Computer (RC) is one special version of RNN, which has been applied to build ENSO prediction model including the study in this article. This study aims to explain the high skill of RC in El Nino prediction theoretically. Based on ideal experiments, the author uses various ZC models in different regimes to generate "observation", and uses RC to learn these data so that the ENSO dynamic characteristics of ZC and the Nino trajectory can be learned. From the results, whether in the subcritical or supercritical regime, RC shows high performance. In addition, through CNOP calculation, the sensitivities of RC and ZC to the initial field are explored with meaningful results. However, it seems to be contrary to the main purpose of the study, and it does not seem to fully explain the reason why RC can produce high El Nino forecasting skills. This is my biggest doubt about this work, and of course it is also the most interesting point. I hope the author can have a more elegant explanation. In addition, the following are some thoughts and suggestions on this work or article:*

**Major comments:**

1. *The results in Figure 2 make me think deeply. It tells us that when training a model, it doesn't mean that the richer the data included, the better the results will be. It seems to be related to the inherent dynamic characteristics of the system. Could you please explain why. For example, in the sub-critical state, why the prediction skill is better when wind field is not included in the training period? Although you attribute*

*it to the sensitivity to wind noise, this is not specific enough. I think it can be discussed in more detail. By the way, I do not understand the sentence in Line 115.*

**Author's Reply:**

In Machine Learning, adding input features does not always enhance performance, as redundant or irrelevant information can degrade model efficiency. In the subcritical regime, where ENSO variability is primarily noise-driven and the system is linearly stable, including surface wind speed anomalies ($\tau_c$) during training may appear beneficial. However, our results show that the impact of $\tau_c$ depends on the forecast horizon. When initialized from ENSO neutral conditions, optimal atmospheric noise patterns can trigger transient growth of perturbations, provided the initial conditions are favorable. Conversely, if a perturbation is already developing, subsequent noise patterns can either reinforce or dampen its evolution. This makes $\tau_c$ particularly useful for predicting short-term variability, as it provides critical information about the external forcing that influences early perturbation dynamics. Accordingly, the Reservoir Computer (RC) achieves better accuracy at shorter lead times (3–6 months) when $\tau_c$ is included. At longer lead times (9–18 months), improved predictive performance requires the model to rely more on the system's internal dynamics rather than the short-term influence of stochastic noise. Including $\tau_c$ during training can lead to overfitting, causing the model to focus excessively on short-term noise patterns instead of learning the internal system dynamics. As a result, model performance deteriorates at longer lead times when $\tau_c$ is included. On line 115, we clarify that instead of directly using zonal wind stress anomalies to train the RC and LR model, we use zonal surface wind speed anomalies as a proxy. These two variables are inherently correlated through the bulk formula, conveying similar information. However, a key distinction arises due to how noise is introduced in the Zebiak and Cane (ZC) model: we introduce stochasticity in the form of random zonal wind stress bursts. This results in random local fluctuations in the zonal wind stress signal that are inherently difficult for the RC and LR models to predict and reproduce. In contrast, the surface wind speed anomaly signal is smoother and more predictable, making it easier for the RC and LR models to learn and generalize effectively.

To illustrate this, Fig. 1 below shows the relationship between zonal surface wind speed anomalies and zonal wind stress anomalies, both normalized by their mean and standard deviation, in the supercritical and subcritical regimes.

**Changes in the Manuscript:**
We will more clearly describe the RC performances in the "RC perfor­mances" section on page 8, focusing on the contribution of the variable $\tau_c$ in the subcritical and supercritical regime. We will better explain why we choose to use surface zonal wind speed anomalies as a proxy for wind stress in the "Reservoir Computer" section.

[Figure]

Figure 1: Relationship between normalized zonal surface wind speed anomalies and zonal wind stress anomalies in the subcritical (a) and supercritical regimes (b).

2. *For the CNOP part, "lead time" in ms is optimization time, right?*

   **Author's Reply:**
   In Fig. 4 and 5 in the "CNOP Analysis" section, the lead time in months on the x-axis corresponds to the optimization time considered during CNOP computation.

   **Changes in the Manuscript:**
   We will clarify that the lead time on the x-axis of Fig. 4 and 5 represents the optimization time considered during CNOP computation.

3. *Some pictures need to be refined. For example, it is recommended that the abscissa and ordinate in Figure A1 should be changed into the format of latitude and longitude coordinates.*

   **Author's Reply:**
   Suggestions will be followed.

   **Changes in the Manuscript:**
   In the revised manuscript, we will refine the figures, including modifying the axes in Fig. A1.

4. *The calculation of CNOP in complicated climate models has always been a major problem. How did you use the gradient-free Cobyla optimization algorithm to solve it? In addition, it is necessary to further verify whether the obtained CNOP is truly the CNOP. It is recommended to add random small perturbations to the obtained CNOP and project it onto the constraint conditions to compare the development of errors, so as to prove that the solution of CNOP is optimal.*

   **Author's Reply:**
   The Constrained Optimization BY Linear Approximation (COBYLA) algorithm is a gradient-free optimization method designed for solving

nonlinear optimization problems. At each iteration, the algorithm constructs linear approximations of the objective function and constraints using linear interpolation at n + 1 points in the space of the optimization variables. The worst-performing point is identified based on the original, not approximated objective function. Using the linear approximations, the algorithm then formulates and solves a linear optimization problem within a small radius around this point to update its value. The COBYLA algorithm is particularly well-suited for nonlinear optimization problems with a relatively small number of variables, especially in cases where computing derivatives is challenging or infeasible. These features make COBYLA an ideal choice for our analysis. Our study compares how the error due to initial uncertainties evolves over time for two different models, the RC and the ZC model. The RC is trained on one-dimensional indices, including the NINO3 index, the mean thermocline depth anomalies in two regions (5°N–5°S, 120°E–180°E in the western Pacific and 5°N–5°S, 180°E–290°E in the eastern Pacific), and the zonal surface wind speed anomalies ($\tau_c$) over the area 5°N–5°S, 145°E–190°E. In contrast, the state vector of the ZC model consists of 2-dimensional fields of sea surface temperature, thermocline depth, oceanic and atmospheric velocities, and atmospheric geopotential. To address these differences in state vector dimensionality and ensure a fair comparison between the RC and ZC models, we have applied a distinct uniform constant perturbation to all the ZC model's SST fields in the NINO3 area, all the thermocline depth fields over the area 5°N-5°S 120°E-180°E and all the thermocline fields over the area 5°N-5°S 180°E-290°E. This has been done to change the mean values over these three areas of a specific quantity. By doing so, we also reduced the number of variables of our optimization problem to three, making the COBYLA algorithm an appropriate choice for our analysis. To validate the estimated Conditional Optimal Nonlinear Perturbations (CNOPs), we didn't use the methodology suggested by the reviewer. Instead, we evaluate error propagation resulting from applying numerous randomly chosen initial perturbations that satisfy the constraint conditions to determine whether the CNOPs correspond to the largest error growth. Furthermore, we verify whether the estimated CNOPs lie on the boundary of the sphere defined by the constraints.

**Changes in the Manuscript:**
In the revised manuscript, we will include a new appendix section providing a detailed description of the COBYLA algorithm. Additionally, we will provide a more detailed description on the validation of the CNOP.

5. *Compared with linear regression, it seems that the advantages of RC are not particularly significant either. What's your view on this issue?*

**Author's Reply:**

While we acknowledge that the RC does not drastically outperform the LR, our results demonstrate a clear advantage in adopting the RC, as its ability to capture nonlinear relationships between input variables, made possible by the use of a nonlinear activation function (the hyperbolic tangent in our study), leads to a consistent performance improvement, particularly in the supercritical regime, where non-linearities play a more prominent role. This is further supported by the fact that in this regime, model performance improves when $\tau_c$ is included during training, highlighting the importance of the nonlinear effects introduced by this variable [1]. These effects are better captured by the RC, whereas the LR can only provide a linear approximation. Furthermore, the relatively small performance gap found in this study can be attributed to the ZC model being a model of intermediate complexity in which ENSO is a weakly nonlinear phenomena (e.g. all wave dynamics in ocean and atmosphere is linear in the model). The data generated from the ZC model exhibit simpler dynamics compared to real-world observations or data from simulations with more complex General Circulation Models (GCMs). In such cases, the performance advantage of the RC over the LR is expected to be more pronounced.

**Changes in the Manuscript:**
In the revised manuscript, we will clarify the difference in performances between the RC and the LR in the "RC performances" section, explaining why the increase in performances is moderate.

6. *In RC, CNOP is not sensitive to the forecast duration, while the opposite is true in ZC (Fig. 5). Why is this the case and what does it imply?*

**Author's Reply:**

Figure 5 shows that the Zebiak-Cane (ZC) model is more sensitive to initial sea surface temperature (SST) perturbations at shorter lead times (3 months) and more sensitive to initial thermocline depth perturbations at longer lead times (6 and 9 months). In contrast, the RC appears to be more sensitive to SST perturbations across all lead times (3 to 9 months). As discussed in the conclusion, thermocline anomalies play a crucial role in error propagation, particularly in the ZC model, where ENSO variability is strongly influenced by the thermocline feedback. The Reservoir, however, effectively reduces sensitivity to initial thermocline perturbations, reducing error propagation.

**Changes in the Manuscript:**
We will include a better clarification of the results in the "CNOP analysis" section on page 11 of the revised manuscript.

7. *Personally, to explain the advantages of RC in ENSO prediction, the key is to focus on the extent to which RC has learned the ENSO dynamics or nonlinear behaviors.*

**Author's Reply:**
In our view, performing short-term forecasts and developing a perfect model emulator capable of capturing the long-term dynamics and statistical properties of a system are fundamentally different tasks, each requiring distinct model architectures, hyperparameter configurations, and evaluation criteria. Assessing how well a Machine Learning model captures a system's dynamics and nonlinear behaviors is more relevant when analyzing its ability to replicate the long-term characteristics and statistics of the system (in the case of ENSO, on a decadal timescale). In recent years, various Machine Learning models have demonstrated strong ENSO forecasting skills on relatively short timescales (up to 21 months) without necessarily capturing all the underlying physical processes. This suggests that their ability to achieve high short-term

predictive skill relies on different factors. Here, we define forecasts spanning a couple of years as "short-term" compared to the decadal timescales required to assess ENSO's long-term behavior. In this study, we focus on the RC model, specifically investigating the hypothesis that its superior predictive performance, particularly its ability to overcome the spring predictability barrier, stems from its capacity to reduce error propagation caused by initial uncertainties. This aligns with the perspective proposed in previous studies [2], where the spring predictability barrier in the ZC model was quantified in terms of sensitivity to initial perturbations.

**Changes in the Manuscript:**
We will make a remark on this in the revised "Summary and Discussion" section.

**References**

[1] Wansuo Duan, Yanshan Yu, Hui Xu, and Peng Zhao. Behaviors of nonlinearities modulating the el niño events induced by optimal precursory disturbances. *Climate Dynamics*, 40(5):1399–1413, March 2013.

[2] Mu Mu, Hui Xu, and Wansuo Duan. A kind of initial errors related to "spring predictability barrier" for el niño events in zebiak-cane model. *Geophysical Research Letters*, 34(3), 2007.

---

## Author Comment (AC3)

**MS-No.:npg-2024-24**

**Title:** Explaining the high skill of Reservoir Computing method in El Niño prediction

**Authors:** Francesco Guardamagna, Claudia E. Wieners and Henk A. Dijkstra

**Point-by-point reply to reviewer #3**

February 9, 2025

We thank the reviewer for their careful reading and for the useful comments on the manuscript.

**Overview**

*This manuscript investigates the prediction skill of a specific type of Recurrent Neural Network, known as Reservoir Computer (RC), in relation to ENSO forecasting. It finds that error propagation in RC is lessened compared to the Zebiak-Cane (ZC) model. While the RC demonstrates high prediction skill (e.g., an ACC greater than 0.6 at 18 months lead time), I believe this manuscript is not suitable for publication for several reasons:*

**Major comments:**

1. *The predictions are not based on real-world data. Both the training and testing datasets are generated from the ZC model, which does not reflect actual observations. It is unclear how well RC performs when predicting realworld events, such as the ENSO events of 2014-2015.*

   **Author's reply:**

   First of all, the RC model's effectiveness in predicting real ENSO events has already been demonstrated in previous studies [2]. Second, we can easily clarify why we only work with data from the Zebiak and Cane (ZC) model rather than real-world observations. The objective of our study is to demonstrate that the Reservoir Computer (RC) can mitigate error propagation resulting from initial conditions perturbations more effectively than a classical dynamical numerical model. This is proposed as a potential explanation for the RC model's high performances in ENSO forecasting and its ability to overcome the Spring

Predictability Barrier problem, which was previously quantified in the ZC model in terms of sensitivity to initial conditions perturbations [3]. Such an analysis and comparison is simply impossible using real-world observations as we do not know the evolution operator of the real-world system and hence cannot determine the CNOP. By focusing on the ZC model data, we analyze the RC model's behavior and learned dynamics within a controlled environment.

**Changes in manuscript:**
No changes in the manuscript needed.

2. *The prediction accuracy of RC is very similar to that achieved by linear regression (LR, as shown in Fig. 2). First, error bars should be included for the LR results. Second, the performance of LR is comparable to that of RC, particularly as indicated by the proximity of the red and blue lines at lead times of 1-9 months.*

**Author's reply:**
In Fig. 2, the yellow and red lines represent the performance of the LR with and without surface wind speed anomalies ($\tau_c$) included during training, respectively. Similarly, the blue and green lines correspond to the performance of the RC with and without $\tau_c$ included during training, respectively. To ensure a fair comparison, the yellow line should be compared with the blue line (LR vs. RC with $\tau_c$ included), and the red line with the green line (LR vs. RC without $\tau_c$ included).

While we acknowledge that the RC does not drastically outperform the LR, our results demonstrate a clear advantage in adopting the RC, as its ability to capture nonlinear relationships between input variables, made possible by the use of a nonlinear activation function (the hyperbolic tangent in our study), leads to a consistent performance improvement, particularly in the supercritical regime, where nonlinearities play a more prominent role. This is further supported by the fact that in this regime, model performance improves when $\tau_c$ is included during training, highlighting the importance of the nonlinear effects introduced by this variable [1]. These effects are better captured by the RC, whereas the LR can only provide a linear approximation.

It is impossible to show error bars for the LR model because, unlike

the RC, the LR does not rely on random weights initialization. The LR will consistently produce the same results for a given training set, so given a specific training set, there is no variability in the LR outputs.

**Changes in manuscript:**
We will better describe the difference between the RC and LR performances in the "RC performances" section.

3. *The influence of wind stress in RC is inconsistent. In some instances, incorporating wind stress enhances ENSO predictions, while in others, it does not. This inconsistency undermines the conclusions drawn, as it does not provide clear insights for real-world predictions, particularly regarding whether ENSO is damped or self-exciting in actual observations.*

**Author's reply:**
We appreciate this critical comment of the reviewer, but our results are actually consistent and show a clear pattern.

In the supercritical regime, the RC consistently performs better across all lead times when $\tau_c$ is included during training. This highlights the importance of the nonlinear effects introduced by this variable, which the RC can efficiently capture through the use of a nonlinear activation function. In the subcritical regime, the Reservoir Computer (RC) achieves higher accuracy at shorter lead times (3–6 months) when $\tau_c$ is included, while at longer lead times (9–18 months), performance improves when $\tau_c$ is excluded. This is because $\tau_c$ plays a crucial role in capturing short-term variability, providing valuable information about the external stochastic forcing that drives the early perturbations dynamics. At longer lead times (9–18 months), improved predictive performance requires the model to rely more on the system's internal dynamics rather than the short-term influence of stochastic noise. Including $\tau_c$ during training can lead to overfitting, causing the model to focus excessively on short-term noise patterns instead of learning the internal system dynamics. As a result, model performance deteriorates at extended lead times when $\tau_c$ is included. These results clearly show how the inclusion of the variable $\tau_c$ influences the RC performances in

the subcritical and supercritical regimes.

Drawing conclusions about the true nature of ENSO is not the objective (and far beyond the scope) of this study.

**Changes in manuscript:**
In the revised manuscript, we will clarify in the "Summary and Discussion" section that the goal of our study is not to draw conclusions about the true nature of ENSO dynamics. Rather, we aim to provide a potential explanation for the RC model's high forecasting performance. We will also better explain the influence of the variable $\tau_c$ in the subcritical and supercritical regimes in the "RC performances" section.

4. *The results from the ZC model raise concerns. For instance, in Fig. A2, the Nino3 index only fluctuates between 0.1 and -0.1.*

**Author's reply:**
Fig. A2 only illustrates the response of the deterministic ZC model to a small initial perturbation applied to the seasonal background state in the subcritical ($r_d < 0.8$) and supercritical ($r_d \geq 0.8$) regimes. In the subcritical regime, the perturbation rapidly decays, and without noise, oscillations cannot occur. In contrast, in the supercritical regime, the perturbation evolves into a stable limit cycle with a period of approximately 4 years. Fig. A2 does not show the actual long term behaviour of the ZC model in the presence of noise, which is depicted in Fig. 1.

**Changes in manuscript:**
No changes in the manuscript needed.

**References**

[1] Wansuo Duan, Yanshan Yu, Hui Xu, and Peng Zhao. Behaviors of nonlinearities modulating the el niño events induced by optimal precursory disturbances. *Climate Dynamics*, 40(5):1399–1413, March 2013.

[2] Forough Hassanibesheli, Jürgen Kurths, and Niklas Boers. Long-term enso prediction with echo-state networks. *Environmental Research: Climate*, 1(1):011002, jul 2022.

[3] Yanshan Yu, Wansuo Duan, Hui Xu, and Mu Mu. Dynamics of nonlinear error growth and season-dependent predictability of el niño events in the zebiak–cane model. *Quarterly Journal of the Royal Meteorological Society*, 135(645):2146–2160, 2009.

---

## Author Comment (AC4)

**MS-No.:npg-2024-24**

**Title:** Explaining the high skill of Reservoir Computing method in El Niño prediction

**Authors:** Francesco Guardamagna, Claudia E. Wieners and Henk A. Dijkstra

**Answer to the comment by Paul Pukite**

February 9, 2025

**Comment:**

Because of the importance of the thermocline in ENSO behavior, the impact of long-period tides in a reduced effective gravity environment has to be included in any predictive analysis. This is particularly appropriate for machine learning, where known tidal data can be straightforwardly included as with any other input. It's obvious from the paper that the concentration focuses on natural responses (see the reproduced Fig.A2(a ) below) which clearly shows the damping characteristic of the perhaps stochastically-selected (via noise) eigenvalue solution to a differential equation.

" This distinction hinges on whether ENSO variability occurs as a sustained oscillation or limit cycle (supercritical) or is a damped oscillation excited by stochastic forcing (subcritical)."

Yet, it's more than likely that ENSO is the result of a forced response to tidal forces, with the annual nonlinear interaction creating an erratic cycling about the approximate 4 year mean period estimated from an index such as NINO3. For the main long-period tidal factors of Mf and Mm, the annually sidebanded periods are calculated at 3.8 and 3.9 years. The complete non-linear solution of the shallow-water Laplace's tidal equations used to model oceanic fluid dynamics is described in [5]. A similar training/validation/test procedure is used for finding an optimal predictive fit as that used in machine learning. The main point in this type of modeling is that predictive analysis can conceivably be made years in advance. The continually forcing of the mixed lunar and annual cycles will create the requisite temporal boundary/guiding conditions to maintain coherence over a long range, much like conventional tides do for sea-level height (SLH) analysis.

**Author's reply:**

Including tidal information as an input variable during the training phase of a Machine Learning (ML) framework could be interesting to investigate. However, previous studies [1], [2], [4] have already demonstrated outstanding ENSO forecasting performance using a combination of sea surface temperature anomalies (SST) and upper ocean heat content anomalies (OHC), or even only using SST anomalies [3]. These findings suggest that SST and OHC anomalies provide enough information for skillful ENSO predictions. Furthermore, our study relies on data generated by the Zebiak and Cane model (ZC), which has been used as a testbed. Since the ZC model does not account for long-period tidal effects, incorporating tidal information during the training of our ML framework would be impossible.

**Changes in manuscript:**

No changes required in the modified manuscript.

**References**

[1] Yoo-Geun Ham, Jeong-Hwan Kim, and Jing-Jia Luo. Deep learning for multi-year ENSO forecasts. *Nature Publishing Group*, pages 1–17, September 2019.

[2] Jie Hu, Bin Weng, Tianqiang Huang, Jianyun Gao, Feng Ye, and Lijun You. Deep residual convolutional neural network combining dropout and transfer learning for enso forecasting. *Geophysical Research Letters*, 48, 12 2021.

[3] Chibuike Chiedozie Ibebuchi and Michael B Richman. Deep learning with autoencoders and LSTM for ENSO forecasting. *Climate Dynamics*, 62(6):5683–5697, June 2024.

[4] Jahnavi Jonnalagadda and Mahdi Hashemi. Long lead enso forecast using an adaptive graph convolutional recurrent neural network. *Engineering Proceedings*, 39(1), 2023.

[5] P. Pukite, D. Coyne, and D. Challou. *Mathematical Geoenergy: Discovery, Depletion, and Renewal*. Geophysical Monograph Series. Wiley, 2019.

---

## Author Response (AR1)

**MS-No.:npg-2024-24**

**Title:** Explaining the high skill of Reservoir Computing method in El Niño prediction **Authors:** Francesco Guardamagna, Claudia E. Wieners and Henk A. Dijkstra

**Point-by-point reply to reviewer #1**

**March 12, 2025**

We thank the reviewer for their careful reading and for the useful comments on the manuscript.

**Overview**

This study focuses on explaining the origins of high forecasting skills of recently emerging deep learning models for ENSO predictions. More specifically, the authors firstly build a skillful Reservoir Computers (RC) model for simulating Zebiak-Cane numerical model, and then investigate and compare the sensitivities on initial perturbations of RC and ZC models (also including an LC model). The authors find RC models are less susceptible to initial perturbations no matter for short and long lead months, which is also the possible reason of weakening the impact of spring predictability barrier (SPB) and extending the effective lead time. In general, this study crafts some novel experiments, and I have the following major questions for the author to answer:

**Major comments:**

1.  $W_{in}$  in Equation (4a) and  $W^{in}$  in the following illustration is not consistent.

**Author's reply:**

We thank the reviewer for pointing out this imprecision.

**Changes in manuscript:**

We have addressed this imprecision in the revised manuscript. Please refer to lines 104–107.

2. What is the value of  $N_x$  in your study, which is quite an important configuration for forecasting skill of RC model.

**Author's reply:**

For both the supercritical and subcritical regimes, as well as for each set of training variables, we determined the optimal hyperparameter sets using a Bayesian search. The search consistently converged on large reservoir dimensions  $N_x$ , with notable differences between the supercritical and subcritical regimes. In the supercritical regime, the optimal reservoir dimension is approximately 400, regardless of whether zonal surface wind speed anomalies ( $\tau_C$ ) are included in the training variable set. In the subcritical regime, the optimal reservoir dimension is larger, with  $N_x = 476$  when  $\tau_C$  is included and  $N_x = 534$  when  $\tau_C$  is excluded.

**Changes in manuscript:**

In the revised manuscript, we have specified the  $N_x$  values used in our study and described the function and impact of this hyperparameter on the Reservoir model (please refer to lines 194–204). Moreover, a summary table reporting the optimal hyperparameter sets for each regime and training variable set has been added to Appendix C.

3. Around line 169, the authors mention that "the initial reservoir state for each prediction was determined using 5 years of data before the time t". Can you explain this expression more to those unfamiliar with RC models?

**Author's reply:**

We agree with the reviewer that this expression may not be clear to those unfamiliar with the Reservoir Computer (RC) model. Discarding the initial x(n) reservoir states for  $0 \le n \le n_{transient}$  before training and evaluation is a standard practice in Reservoir Computing. This step is necessary to mitigate the impact of initial transients caused by the arbitrary initialization of the reservoir state, which is typically set to x(0) = 0 or initialized randomly. In our case, the reservoir state was initialized as x(0) = 0. This initialization creates an artificial starting state that is unlikely to recur once the reservoir dynamics stabilize. A warmup period is therefore introduced to allow the reservoir to reach a stable dynamical regime before training or inference. The length of the warmup period depends on the reservoir's memory capacity and the specific learning task. Based on our experiments, a warmup period of 5 years is sufficient to stabilize the reservoir dynamics and eliminate the effects of initial transients. As a result, before inference, the reservoir states corresponding to the 5 years preceding the initial time step t(n) (the starting point of our forecast) are discarded. In our notation, this means discarding x(n) reservoir states for  $t(n) - n_{transient} \leq n \leq t(n)$ , where  $n_{transient} = 180$ , given our 10-days time step.

**Changes in manuscript:**

In the revised manuscript, we have clarified why a warm-up period is necessary when working with a Reservoir Computer model in the 'Training and Validation' section. Please refer to lines 180-189.

4. Around line 224, the authors mention that "This result aligns with expectations, as non-linearities play a more important role in the supercritical regime". I am wondering is it true that the supercritical regime exhibits more nonlinear than subcritical regime, which favors the performance of the RC model? What's the relationship between nonlinearity of regime and RC model performance?

**Author's reply:**

ENSO can be described by two different theoretical frameworks. According to one perspective, it is a stable (damped) mode sustained primarily by random atmospheric noise (subcritical regime). Alternatively, it can be viewed as a self-sustained oscillatory mode (supercritical regime). In the latter scenario, nonlinearity is essential in modulating ENSO behavior. In the Zebiak and Cane (ZC) model, nonlinearities come from three main sources: heat advection, wind stress anomalies, and subsurface water temperature variations [1].

The Reservoir Computing (RC) model can capture complex nonlinear relationships between input variables by employing a nonlinear activation function (in our study, the hyperbolic tangent). In contrast, a Linear Regressor (LR) can only estimate linear relationships between input variables. Consequently, the performance gap between the RC model and the LR is expected to be more pronounced in the supercritical regime, where nonlinear effects are more important.

**Changes in manuscript:**

In the revised manuscript, in the "Zebiak and Cane (ZC) model" section, we have improved our explanation of why nonlinearities are more influential in the supercritical regime. We now specify the three main sources of these nonlinearities and cite [1], which assesses their relative importance. Please refer to lines 72–76. In the "Reservoir Computer" section, we clarify why the RC model is better suited to capture complex nonlinear relationships between input variables than a simple LR. Please refer to lines 101–104. Furthermore, in the "RC performances" section, we explain why including the variable  $\tau_C$  during training the RC model's performance increase in the supercritical regime, underscoring the crucial role of the nonlinear effects introduced by this variable. Please refer to lines 227–232.

5. Around line 248, do the authors use values of the CNOP objective function to assess whether the model is susceptible?

**Author's reply:**

For both the RC and the ZC model, we used the Cobyla optimization algorithm to maximize the CNOP objective function and estimate the optimal initial perturbations. The resulting maximal error growth, calculated for a specific lead time, was then taken as our measure of the model's sensitivity to initial perturbations.

**Changes in manuscript:**

In the revised manuscript we have better specify this point in the "CNOP analysis" section. Please refer to lines 291-294.

6. For figure 5, why the CNOP results for ZC model is quite symmetrical while the CNOP results for RC models are usually biased?

**Author's reply:**

The ZC model's optimal initial perturbations exhibit a notably symmetrical distribution, evident in both SST perturbations at shorter lead times (3 months) and thermocline depth perturbations at longer lead times (6 and 9 months). This symmetry suggests that the model is sensitive to both negative and positive initial perturbations, depending on the specific event.

In contrast, the RC generally demonstrates greater sensitivity to initial SST perturbations across both shorter and longer lead times, and the distribution of these optimal initial SST perturbations consistently shows a clear preference for either positive or negative values.

To better understand the origins of these differences, we have divided the initial conditions considered in the "CNOP analysis" section, spanning the months [March,April,May] into three categories (positive, negative, and neutral) based on the initial eastern Pacific sea surface temperature anomalies (SSTA). This categorization allowed us to compare the CNOPs behavior for each category across both the ZC model (in the subcritical and supercritical regimes) and the RC model (in both regimes, with and without the inclusion of the variable  $\tau_C$  during training).

**Changes in manuscript:**

In the revised manuscript, we have incorporated the results of this new analysis in the "CNOPs analysis" section. Please refer to lines 346–424. Additionally, Appendix D now includes two plots that illustrate the distribution of CNOPs for the various initial condition categories. These plots cover both the ZC model (in the subcritical and supercritical regimes) and the RC model (in the subcritical and supercritical regimes, with and without the inclusion of the variable  $\tau_C$ during training).

7. Around line 297, the authors mention that the inclusion or exclusion of wind speed anomalies has a large effect on the different variables of CNOP in the RC models for the subcritical regime. I am wondering why this is not obvious in the RC models for the supercritical regime?

**Author's reply:**

In the subcritical regime, ENSO variability is primarily sustained by atmospheric noise, introduced as stochastic wind stress forcing. This noise influences the thermocline slope, activating mechanisms that lead to the development of perturbations. When the variable  $\tau_c$  is included during training, the RC explicitly learns the relationship between wind anomalies and thermocline adjustments, and the state of the surface winds is provided as an initial condition. Consequently, smaller thermocline perturbations can be amplified by wind anomalies, leading to larger deviations from the reference trajectory. In contrast, when  $\tau_c$  is not included, the RC only learns the direct relationship between SST and thermocline depth anomalies without explicit knowledge of how wind anomalies influence thermocline slope adjustments. As a result, in the absence of wind-forcing information, a larger initial thermocline perturbation is required to generate significant error propagation over time.

In the supercritical regime, thermocline depth perturbations are not purely activated by atmospheric noise but also emerge due to the internal instability of the system. The influence of stochastic atmospheric noise is weaker than in the subcritical case. This means that even if the model does not explicitly account for the effect of wind forcing on the thermocline slope, strong initial thermocline depth anomalies are not needed to maximize error propagation.

However, even in the supercritical regime, when  $\tau_c$  is not included, the RC remains more sensitive to initial thermocline perturbations at longer lead times than when  $\tau_c$  is included, but the difference is much less pronounced than in the subcritical regime.

In every case, the Reservoir Computer is consistently less sensitive to thermocline depth perturbations at longer lead times compared to the Zebiak and Cane model. This suggests that the RC effectively mitigates error propagation from thermocline perturbations.

**Changes in manuscript:**

In the revised manuscript, in the "CNOP analysis" section, we have provided a clearer explanation of why the inclusion of the variable  $\tau_C$ has a greater impact on the RC model's optimal initial perturbations in the subcritical regime than in the supercritical regime, connecting this explanation with the new analysis provided for the previous comment about the symmetrical distribution of the ZC model CNOPs and the more biased distribution of the RC model's CNOPs. Please refer to lines 389–424 for further details.

8. I have noticed there is another similar study that revealing the initial perturbations of SST for ENSO predictions in AI model (https://doi.org/10.1002/qj.4882). Maybe this is a more comprehensive way to detecting the detailed patterns and physical variables of initial perturbations related to SPB from index models (such as RC models used in this study) to spatial models.

**Author's reply:**

We thank the reviewer for bringing to our attention a study similar to ours that was not cited in our manuscript. Although the analysis by Qin et al. [2] bears similarities to our experiments, there are substantial differences that make both studies valuable:

- (a) In [2], the GFDL CM2p1 dynamical numerical model is used solely to validate the optimal initial perturbations computed for the Deep Learning model employed in their study. However, they do not compute the optimal initial perturbations for the GFDL CM2p1 model itself. As a result, their analysis does not explore whether the GFDL CM2p1 and Deep Learning models exhibit similar optimal initial perturbations or how optimal initial errors propagate in both models. This comparison is a central aspect of our study.
- (b) In [2], the CNOP objective function for the deep-learning model is optimized using automatic differentiation, a feature available in most modern deep-learning frameworks, such as PyTorch and TensorFlow. However, this approach does not apply to dynamical numerical models since these models are not implemented using modern deep-learning frameworks. Moreover, for these models,

computing or approximating gradients with respect to their outputs is often highly complex or practically infeasible, making it challenging to apply gradient-based optimization techniques. To enable a meaningful comparison of how optimal initial perturbations evolve in both machine learning and numerical models, a gradient-free optimization method, like the Cobyla algorithm employed in our study, is essential.

These differences in the analysis performed and the methods adopted underscore the complementary contributions of our study and that in [2].

**Changes in manuscript:**

In the revised manuscript, in the "Summary and Discussion" section, we reference [2] to discuss their interesting results and underline the key differences between their study and ours. Please refer to lines 433–445.

9. The AI model appears to be less sensitive to the initial perturbations, or the initial perturbations do not grow as fast as those in numerical models, which is the reason for the higher skill of the AI model. Similar conclusions are also obtained in another similar study (https://doi.org/10.1029/2023GL105747). Can the authors discuss the pros and cons of this characteristics? I think this will be significantly valuable for the future modelling, as well as further understanding, of earth system.

**Author's reply:**

We agree with the referee that discussing the pros and cons of this characteristic of AI models compared to dynamical models will be highly valuable.

**Changes in manuscript:**

In the "Summary and Discussion" section of the revised manuscript, we discussed the pros and cons of this characteristic of AI models, referring

to the interesting results presented by Selz et al. in [3]. Please refer to lines 467-473.

Authors' comments: Due to minor imprecisions in our initial results, we have updated two entries in Tables D1 and D2 in Appendix D. Consequently, Fig. D1 (a)–(d) has been revised to reflect these corrections; the differences from the previous version are minimal. Additionally, Fig. 6(b) and Fig. 5(c)–(d) have been updated, with the observed variations being negligible and not affecting our final conclusions. The corrected results have been uploaded to Zotero, and the new link is provided in the code availability section.

**References**

- Wansuo Duan, Yanshan Yu, Hui Xu, and Peng Zhao. Behaviors of nonlinearities modulating the el niño events induced by optimal precursory disturbances. *Climate Dynamics*, 40(5):1399–1413, March 2013.
- [2] Bo Qin, Zeyun Yang, Mu Mu, Yuntao Wei, Yuehan Cui, Xianghui Fang, Guokun Dai, and Shijin Yuan. The first kind of predictability problem of el niño predictions in a multivariate coupled data-driven model. *Quarterly Journal of the Royal Meteorological Society*, 150(765):5452–5471, 2024.
- [3] T. Selz and G. C. Craig. Can artificial intelligence-based weather prediction models simulate the butterfly effect? *Geophysical Research Letters*, 50(20):e2023GL105747, 2023. e2023GL105747 2023GL105747.

**MS-No.:npg-2024-24**

**Title:** Explaining the high skill of Reservoir Computing method in El Niño prediction **Authors:** Francesco Guardamagna, Claudia E. Wieners and Henk A. Dijkstra

**Point-by-point reply to reviewer #2**

March 12, 2025

We thank the reviewer for their careful reading and for the useful comments on the manuscript.

**Overview**

Reservoir Computer (RC) is one special version of RNN, which has been applied to build ENSO prediction model including the study in this article. This study aims to explain the high skill of RC in El Nino prediction theoretically. Based on ideal experiments, the author uses various ZC models in different regimes to generate "observation", and uses RC to learn these data so that the ENSO dynamic characteristics of ZC and the Nino trajectory can be learned. From the results, whether in the subcritical or supercritical regime, RC shows high performance. In addition, through CNOP calculation, the sensitivities of RC and ZC to the initial field are explored with meaningful results. However, it seems to be contrary to the main purpose of the study, and it does not seem to fully explain the reason why RC can produce high El Nino forecasting skills. This is my biggest doubt about this work, and of course it is also the most interesting point. I hope the author can have a more elegant explanation. In addition, the following are some thoughts and suggestions on this work or article:

**Major comments:**

1. The results in Figure 2 make me think deeply. It tells us that when training a model, it doesn't mean that the richer the data included, the better the results will be. It seems to be related to the inherent dynamic characteristics of the system. Could you please explain why. For example, in the sub-critical state, why the prediction skill is better when wind field is not included in the training period? Although you attribute it to the sensitivity to wind noise, this is not specific enough. I think it can be discussed in more detail. By the way, I do not understand the sentence in Line 115.

**Author's Reply:**

In Machine Learning, adding input features does not always enhance performance, as redundant or irrelevant information can degrade model efficiency. In the subcritical regime, where ENSO variability is primarily noise-driven and the system is linearly stable, including surface wind speed anomalies  $(\tau_c)$  during training may appear beneficial. However, our results show that the impact of  $\tau_c$  depends on the forecast horizon. When initialized from ENSO neutral conditions, optimal atmospheric noise patterns can trigger transient growth of perturbations, provided the initial conditions are favorable. Conversely, if a perturbation is already developing, subsequent noise patterns can either reinforce or dampen its evolution. This makes  $\tau_c$  particularly useful for predicting short-term variability, as it provides critical information about the external forcing that influences early perturbation dynamics. Accordingly, the Reservoir Computer (RC) achieves better accuracy at shorter lead times (3–6 months) when  $\tau_c$  is included. At longer lead times (9–18 months), improved predictive performance requires the model to rely more on the system's internal dynamics rather than the short-term influence of stochastic noise. Including  $\tau_c$  during training can lead to overfitting, causing the model to focus excessively on short-term noise patterns instead of learning the internal system dynamics. As a result, model performance deteriorates at longer lead times when  $\tau_c$  is included. On line 115, we clarify that instead of directly using zonal wind stress anomalies to train the RC and LR model, we use zonal surface wind speed anomalies as a proxy. These two variables are inherently correlated through the bulk formula, conveying similar information. However, a key distinction arises due to how noise is introduced in the Zebiak and Cane (ZC) model: we introduce stochasticity in the form of random zonal wind stress bursts. This results in random local fluctuations in the zonal wind stress signal that are inherently difficult for the RC and LR models to predict and reproduce. In contrast, the surface wind speed anomaly signal is smoother and more predictable, making it easier for the RC and LR models to learn and generalize effectively.

To illustrate this, Fig. 1 below shows the relationship between zonal surface wind speed anomalies and zonal wind stress anomalies, both normalized by their mean and standard deviation, in the supercritical and subcritical regimes.

**Changes in the Manuscript:**

In the revised manuscript, in the "RC performances" section, we provide a clearer explanation of how the variable  $\tau_C$  contributes to RC performance in both the subcritical and supercritical regimes. Please refer to lines 217–232 on page 9. Additionally, in the "Reservoir Computer" section, we better explain why we use surface zonal wind speed anomalies as a proxy for wind stress. Please refer to lines 120–126.

---

## Author Response (AR2)

**MS-No.:npg-2024-24**

**Title:** Explaining the high skill of Reservoir Computing method in El Niño prediction

**Authors:** Francesco Guardamagna, Claudia E. Wieners and Henk A. Dijkstra

**Reply to the editor**

April 4, 2025

**Private note**

The authors could have a discussion in the section "Summary and discussion" on the comment provided by the report #2 in the second round of review. Actually, an RC model trained by realistic data can also calculate the CNOP, whose evolution can be compared with that obtained by the RC model trained by the ZC model output. Such comparison may make you results much robust.

**Author's Reply:**

Suggestion followed.

**Changes in the Manuscript:**

In the "Summary and Discussion" section (lines 442–455), we now mention the possibility of applying the CNOP framework to machine learning models trained with real observations, clarifying why this type of analysis can be useful for ENSO prediction.